# Universal neural networks for real-time earthquake early warning trained with generalized earthquakes
Xiong Zhang [1,2] ✉ & Miao Zhang [3]

Deep learning enhances earthquake monitoring capabilities by mining seismic waveforms directly. However, current neural networks, trained within specific areas, face challenges in generalizing to diverse regions. Here, we employ a data recombination method to create generalized earthquakes occurring at any location with arbitrary station distributions for neural network training. The trained models can then be applied universally with different monitoring setups for earthquake detection and parameter evaluation from continuous seismic waveform streams. This allows real-time Earthquake Early Warning (EEW) to be initiated at the very early stages of an occurring earthquake. When applied to substantial earthquake sequences across Japan and California (US), our models reliably report most earthquake locations and magnitudes within 4 seconds of the initial P-wave arrival, with mean errors of 2.6-7.3 km and 0.05-0.32, respectively. The generalized neural networks facilitate global applications of real-time EEW, eliminating complex empirical configurations typically required by traditional methods.

Earthquake monitoring is a primary task in seismology, and reporting earthquake parameters in real time has long been a critical effort for earthquake early warning (EEW)[1–4]. The current EEW systems typically require a few seconds to 1 min after an earthquake occurs to issue warning information to public[3]. The systems commonly consist of several modules, including data processing, source parameter evaluation, and alert filtering, used to report earthquake alarms based on continuous waveform streams[5–7]. The alarm is triggered if specific conditions are met, such as distinguishing teleseismic events, triggering a certain number of stations, reaching a certain magnitude, and meeting a percentage threshold of triggered stations (e.g., 40%)[5,7]. Complex empirical threshold value settings are involved in each processing step, making it a challenge to define the optimal alert criteria in EEW systems. Implementing overly strict criteria, such as requiring too many or a large number of stations to trigger, can negatively impact the real-time efficiency of EEW systems, while loose criteria can result in false alarms[5]. For EEW by single stations, traditional methods typically use the initial few seconds of P waves to estimate the maximum amplitude of the displacement or period parameter for onsite warning[8–10]. To avoid false alarms, traditional EEW practices often require one or more additional triggered stations to confirm the alert[11,12]. With the development of deep learning, trained neural networks can maximize the information extracted from the initial P waves of single stations, efficiently detecting and predicting

earthquake parameters[13]. For regional EEW by network stations (e.g., ElarmS), earthquake parameter determination usually requires data from at least four triggered stations to ensure accuracy[5–7]. However, in real applications, the time delay for issuing a warning may be even longer due to malfunctions in stations or system delays[6,14]. To accurately determine the magnitudes of large earthquakes, more P waves may be required due to the long source time function of the rupture process[15,16]. This makes the waveforms of every second particularly valuable for EEW. Therefore, an efficient real-time monitoring algorithm should not only be computationally fast in a concise way without extensive empirical configurations but also be able to solve the earthquake parameters by comprehensively using limited data available from stations at the early stage of the earthquake occurring.

Traditional travel time-based earthquake monitoring involves many steps, including earthquake detection, phase picking, phase association, earthquake location, and magnitude evaluation[17–26]. With machine learning applied in some steps, the automatic construction of earthquake catalogs is now feasible without the need for manual intervention[27–29]. Especially, the development of various neural networks for phase detection and association, along with the enhanced generalization ability of these networks[30–32], significantly improves our capacity to monitor and understand seismic activity. However, as the workflow involves many steps, an error in either of

[1]Engineering Research Center for Seismic Disaster Prevention and Engineering Geological Disaster Detection of Jiangxi Province, East China University of Technology, Nanchang, Jiangxi, China. [2]Shanghai Sheshan National Geophysical Observatory, Shanghai, China. [3]Department of Earth and Environmental Sciences, Dalhousie University, Halifax, NS, Canada. ✉e-mail: zxiong@mail.ustc.edu.cn

the steps would potentially affect the final location and magnitude results, and they require more time to receive and analyze the earthquake signals. Unlike the automatic construction of earthquake catalogs, EEW systems require the rapid reporting of earthquake parameters at the very early stages of occurrence, often with limited information available from network stations. This underscores the significance of leveraging the full available information recorded by stations, rather than relying solely on predefined partial features such as arrival times and amplitudes. Even the signals and noises in the not-yet-triggered stations prove to be valuable, suggesting that the earthquake location solution should be far from these stations[33,34].

Another effort to monitor earthquakes without human intervention involves direct data mining from waveforms to detect earthquakes and evaluate parameters using deep learning techniques[35–38], bypassing many intermediate steps that require complex empirical settings such as various threshold values and alert criteria. This approach utilizes waveform features more than phase picks. For single-station monitoring, Perol et al.[38] utilize CNN to detect earthquakes and classify earthquake locations into several blocks within a specified region[38]. Moreover, the epicentral distance, back-azimuth, and magnitude can be determined by combining properly extracted input waveform features with well-designed neural networks[13,35–37]. However, the back-azimuth, which relies on phase particle motion, may be affected by the noise level of the data. To obtain more accurate earthquake parameters, a network of stations may be necessary to provide sufficient constraints for neural network learning[39–46]. Zhang et al.[41] demonstrate that the fully convolutional neural network is efficient in mapping the waveforms of fixed network stations to the earthquake location labeled by the 3D Gaussian distribution function[41]. van den Ende et al.[42] utilize a graph neural network to estimate the earthquake location and magnitude by incorporating the spatial information of the stations into the neural network input[42]. The waveform features from each station are extracted independently and then combined by a component of multi-layer conception; thus the output is not sensitive to the station order. Nevertheless, the generalization of monitoring neural networks remains a challenge due to the diverse geometric distribution of stations or geological structures. These methods commonly require transfer learning when applied to new regions[39–46]. Furthermore, earthquake monitoring for early warning requires neural networks to handle both triggered and not-yet-triggered stations from the outset of an earthquake event[33,34], adding complexity to neural network learning. Zhang et al.[44] extend the fully convolutional neural network to early warning applications, showcasing its capability to determine source parameters at the onset of an earthquake[44]. However, the neural network's application is restricted to the same region and station distribution as the training set.

In this study, we develop data augmentation methods to train generalized neural networks capable of universal real-time earthquake monitoring and early warning in diverse regions with different station distributions. Despite the abundance of seismic datasets shared online within the seismology community, such as STEAD[47], INSTANCE[48], and DiTing[49], it remains a challenge to collect a sufficient number of training earthquakes with diverse station distributions and monitoring settings. In our application, we utilize a recombination method of station seismograms to create numerous generalized training earthquakes that occurred at any location within the study area with arbitrary station distributions. The trained neural networks are applied across diverse regions for universal real-time earthquake early warning, extracting features from limited available data at the early stages of earthquake occurrence, without complex empirical settings as required in traditional methods.

## Results

### Network models and generalized earthquakes
We have designed three neural networks for earthquake detection, location, and magnitude estimation, with seismic data streams continuously fed into the network models (Fig. 1). The earthquakes are simultaneously detected and located with waveforms from multiple stations, and then the neural network estimates the earthquake magnitude for each station when an

earthquake is detected. Our models are built upon fully convolutional networks with detailed configurations described in the "Data and methods" section. The outputs are labeled using 1D or 3D Gaussian distributions to represent the detected P arrival, location, and magnitude. For the location network, we incorporate station XY coordinates as two channels of the network input and sort the input data by station X and Y coordinates, ensuring that a specified earthquake corresponds to a unique input with a well-defined station order. The 3D Gaussian distribution outputted by the neural network represents an earthquake location area of 50 km × 100 km horizontally, while the station distribution covers the earthquake location range within a broader region of 82 km × 110 km. Therefore, in a typical monitoring setting, we assume that the stations are distributed within a range of 0–82 in the X direction and 0–100 km in the Y direction, while the earthquakes occur within the monitoring area of 16–66 km in the X direction, 0–100 km in the Y direction, and 0–20 km in the Z direction. To train the detection and location network, we collect 94,586 single-station seismograms from real earthquakes that occurred in Italy, Oklahoma (US), and Southern California (US) (Fig. 2). These seismograms are grouped according to epicentral distance and earthquake depth to form a base training dataset. To ensure a universal neural network, the key technique involves the production of training samples by recombining single-station seismograms in the base dataset to create generalized earthquakes occurring within the study area with arbitrary station distributions (Fig. 3). Similar to the 1D layered velocity model approximation used in common location methods[23,50], we adopt the assumption that seismograms with the same epicentral distance and depth across various global areas exhibit similarity in phase arrival times. This assumption enables us to recombine collected seismograms and generate training earthquakes in arbitrary monitoring areas. To generate a training sample, we randomly select an earthquake location $(s_x, s_y, s_z)$ within the earthquake range and 4–12 station locations within the station range. We then calculate the epicentral distance $r$, azimuth $\varphi$, and randomly select a three-component waveform with epicentral distance of $r$ and depth of $s_z$ from the grouped base dataset for a synthetic station. Figure 3a illustrates a representative generalized training earthquake monitored by four stations with waveforms from two different real earthquakes in the base dataset. The E and N components of the waveforms $(d_E, d_N)^T$ are rotated to the new azimuth direction centered at the generalized earthquake location by using the equations:

$$\begin{pmatrix} d'_E \\ d'_N \end{pmatrix} = \begin{pmatrix} \cos(\delta) & \sin(\delta) \\ -\sin(\delta) & \cos(\delta) \end{pmatrix} \begin{pmatrix} d_E \\ d_N \end{pmatrix} \tag{1}$$

where $\delta$ is the difference between the original azimuth of the waveform in the base dataset and new azimuth $\varphi$ of the generalized earthquake. This rotation ensures that the generalized training samples retain the azimuth features in the waveform data. Since the generalized earthquake's seismograms may be from actual earthquakes with different magnitudes, we filter the three-component seismograms within a frequency range of 2–8 Hz and normalize their amplitudes to an equivalent magnitude to accommodate inconsistent amplitudes. The single-station waveforms are scaled to the amplitude of $A$ by using Hutton and Boore's empirical equation[51]: $\log(A) = M_L - 1.110 \log(r/100) - 0.00189(r - 100) - 3.0$ for each station. We randomly recombine the seismograms to create 355,001 earthquakes within the monitoring areas for detection and location network training. Additionally, we generate 2000 earthquakes outside the monitoring areas to enhance the network's ability to identify abnormal events, which are labeled as zeros. To simulate the real-time EEW process, we shift the waveforms of the generalized earthquakes and vary the length of effective signals in the current time window (30 s) to generate training samples when only a subset of stations is triggered (see "Data and methods" section). The generalized earthquake data contain the crucial features related to earthquake location, although we omit many other physical factors, such as the focal mechanism and complex geological structures across various regions. The neural networks, trained with the generalized earthquakes, are subsequently applied to various regions (Fig. 2).

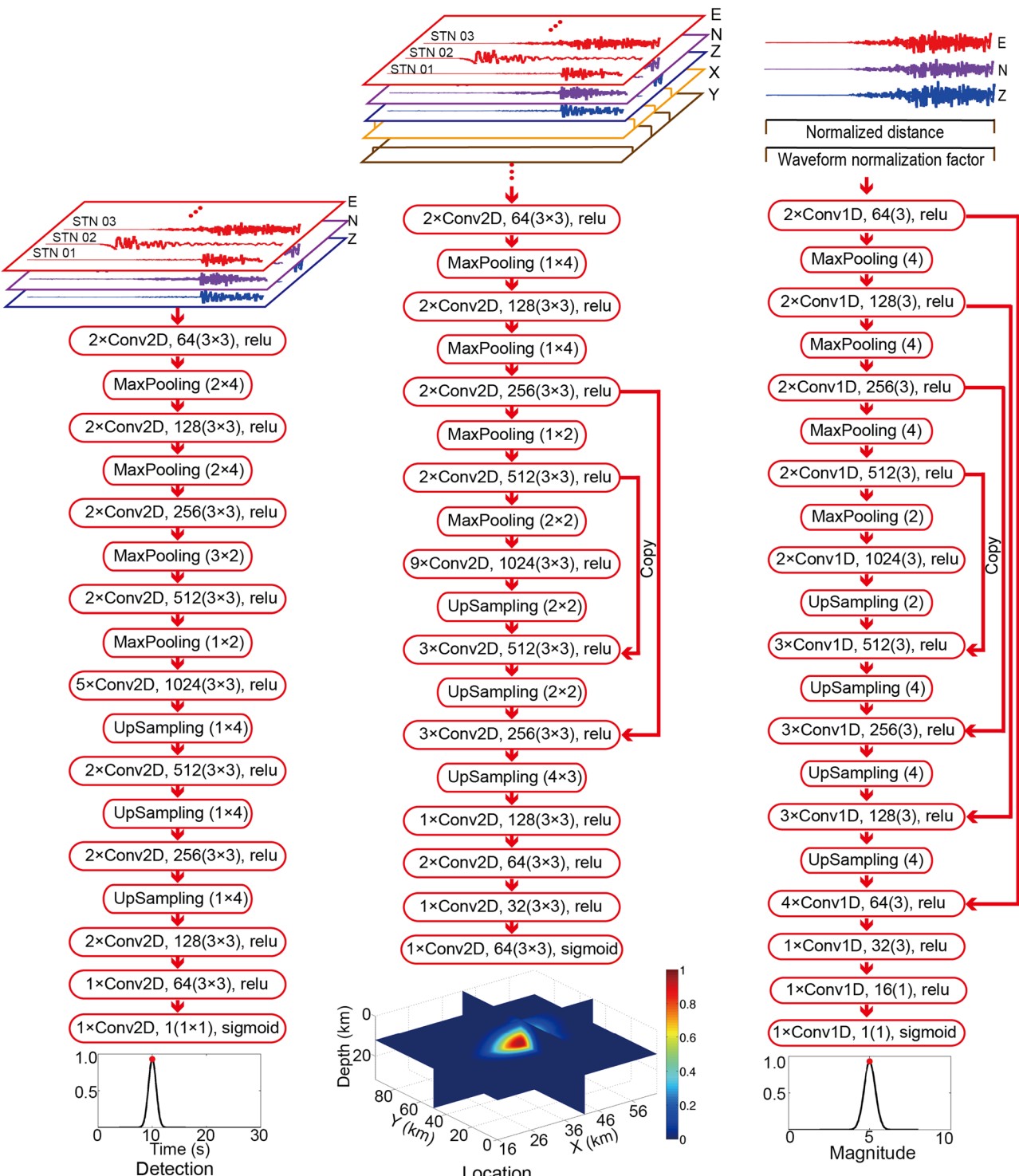

**Fig. 1 | Inputs, outputs, and neural networks.** The three-component waveforms or station XY locations are fed into the neural networks, and the outputs are labeled as 3D or 1D Gaussian distributions to represent the earthquake parameters. The convolutional layers read as a number of kernels (kernel size).

## Pseudo real-time monitoring of the main shocks in Osaka, Japan, and Ridgecrest, US

We apply the trained neural networks to the main shock (M 6.1) that occurred on 18 June 2018 in Osaka, Japan, and the main shock (M 6.4) that occurred on 4 July 2019 in Ridgecrest, US (Fig. 4). The two earthquakes were monitored by 12 stations with different spatial distributions. To simulate real-time monitoring, the continuous waveforms from the 12 stations are fed into the neural network with a 30 s truncating window and 0.5 s time interval along with the station locations.

The detection and location networks output the 1D and 3D probability density functions (PDFs) with Gaussian distributions simultaneously. The maximum PDFs surpassing the given threshold values indicate an earthquake detected and located in the input time window. We set the PDF threshold values to 0.7 and 0.6 after analyzing the predicted results by using some testing samples and found that these values are suitable for monitoring in different areas. The first triggered P arrival and predicted earthquake location are identified by the maximum values in the 1D and 3D PDFs. The epicentral distances for the monitoring stations

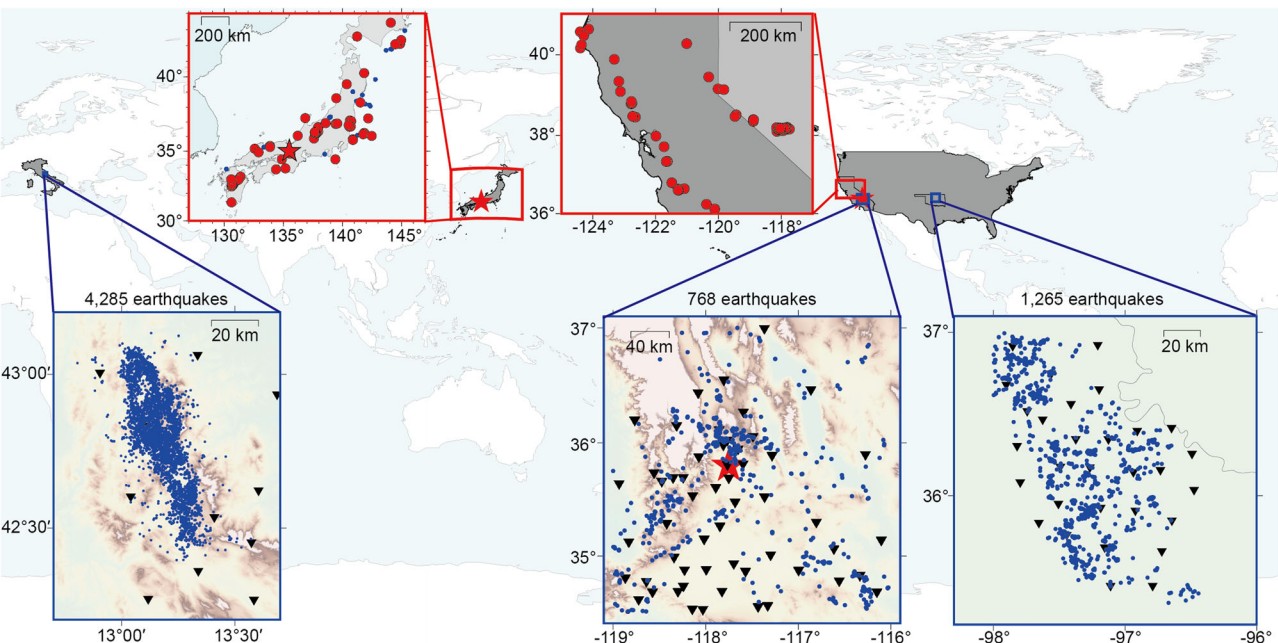

**Fig. 2 | The training data and testing sites.** The blue dots indicate the collected training earthquakes; the triangles represent the recording stations. The red stars denote the testing sites for earthquake sequences in Osaka, Japan, and Ridgecrest, US, while the red rectangles indicate larger testing areas including 139 relatively large earthquakes (red dots) across Japan and Northern California (US).

are calculated by using the predicted earthquake location. The origin time is estimated by subtracting the travel time from the first triggered P arrival time, and then the theoretical P arrival times of all the stations are calculated. If the theoretical P arrival time falls within the current time window, it indicates that the corresponding station is triggered. The magnitude neural network predicts the magnitude by taking the epicentral distance, normalized waveforms, and the normalization factor of each single station as inputs, and outputs the 1D magnitude PDF for each triggered station. The predicted magnitudes are considered robust if the maximum PDFs exceed 0.6 and the earthquake signals last for more than 2 s (estimated by the difference between the current time and the theoretical P arrival time). The final magnitude of the detected earthquake is the mean of the triggered stations.

The neural networks detect the earthquake signals and predict the earthquake parameters instantly with continuous waveform input (Supplementary Movies 1 and 2). In Fig. 4, we present monitoring snapshots for the main shocks in Osaka, Japan, and Ridgecrest, US, captured at the 4th and 15th seconds after the first triggered station. At the 4th second, the epicenter errors are 3.9 km and 3.2 km with maximum location PDFs of 0.698 and 0.665 for the Osaka and Ridgecrest earthquakes, respectively. The magnitudes are 5.5 and 5.8, calculated from the means of 3 and 5 triggered stations meeting the criteria of maximum PDFs exceeding threshold values and earthquake signals lasting over 2 s. The epicenter errors remain similar to the results obtained at the 15th second, even though the latter results are evaluated with more data, however, the magnitudes improve to 6.1 and 6.7 respectively. The results imply that the monitoring system can reliably report earthquake parameters at the 4th second after the earliest P arrival. Although the magnitudes are underestimated with errors of 0.6 for both examples at the 4th second, the estimates still indicate that they are relatively large earthquakes. In fact, the main shocks could be detected and located as early as 2.2 s and 2.7 s, respectively, although the magnitudes are underestimated (M 4.5 and M 5.4), as shown in the Supplementary Fig. 1. This indicates that the first alarms may be reported even earlier if the predicted magnitudes satisfied the lowest magnitude requirements.

## Robustness analysis for the monitoring in Osaka (Japan) and Ridgecrest (US)

We apply the neural networks to 528 relatively large earthquakes and the continuous records of one day's duration in both Osaka, Japan, and Ridgecrest, US to test the robustness of the network models (see Fig. 5). The continuous waveforms are fed into the neural network, and the detection and location threshold values are set to 0.7 and 0.6. If both PDF values are satisfied, the predicted results enter the shortlist of an earthquake, and the input windows with similar origin times and similar locations are considered as the same event. We remove the duplicated events by choosing the result with the best location PDF and the effective signals occur within the range of 15~20 s, considering a reasonable length of waveforms for better parameter accuracy. We detect and locate 126 events on June 18, 2018, in Osaka, Japan, and 389 events on July 4, 2019, in Ridgecrest, US, distributed around the main shocks (Fig. 5). Although the neural networks are trained with M ≥ 2.5 earthquakes, they can also detect and locate earthquakes smaller than M 2.5 (Fig. 5e, f). The M > 3.0 earthquakes predicted by the network models are all present in the original catalog, indicating no false alarms for M > 3.0 earthquakes for both regions. Conversely, all M > 3.0 earthquakes in the original catalog also appear in the network model's catalog for Japan data. However, for the Ridgecrest data, the network model's catalog is missing 15 out of 106 earthquakes with M > 3.0, and 1 out of 16 earthquakes with M > 4.0. This discrepancy is likely due to the high frequency of earthquakes occurring in close temporal proximity as shown in Supplementary Fig. 2. To assess the accuracy of earthquake parameters predicted by the generalized neural networks, we utilize the waveforms of 179 events (M > 2.0) occurred from 16 Jan to 25 Dec 2018, in Osaka, Japan and 349 events (M > 3.5) occurred from 4 July to 16 Nov 2019, in Ridgecrest, US as inputs. The monitoring stations are the same as in the continuous data tests of the main shocks; however, some of the stations may not work or the data are missing for some events in one year range. The epicenter errors for the Osaka and Ridgecrest earthquakes are 2.6 km and 4.5 km (Fig. 5a, b), respectively, and the magnitude errors are 0.08 and 0.11, respectively (Fig. 5c, d). The results show that the earthquake parameters can be evaluated without any human intervention and

**Fig. 3 | Data recombination for creating generalized training earthquakes. a** A typical generalized earthquake produced by recombining seismograms from two real earthquakes; the earthquake locations are represented by stars, and monitoring stations with seismograms are shown as triangles. **b** The base dataset consisted of 94,586 seismograms grouped by epicentral distance and earthquake depth. Four single-station seismograms (S1, S2, S3, S4) in (**a**) are extracted from the base dataset and marked as white dots. **c** The magnitude statistics of the base dataset for detection and location network training.

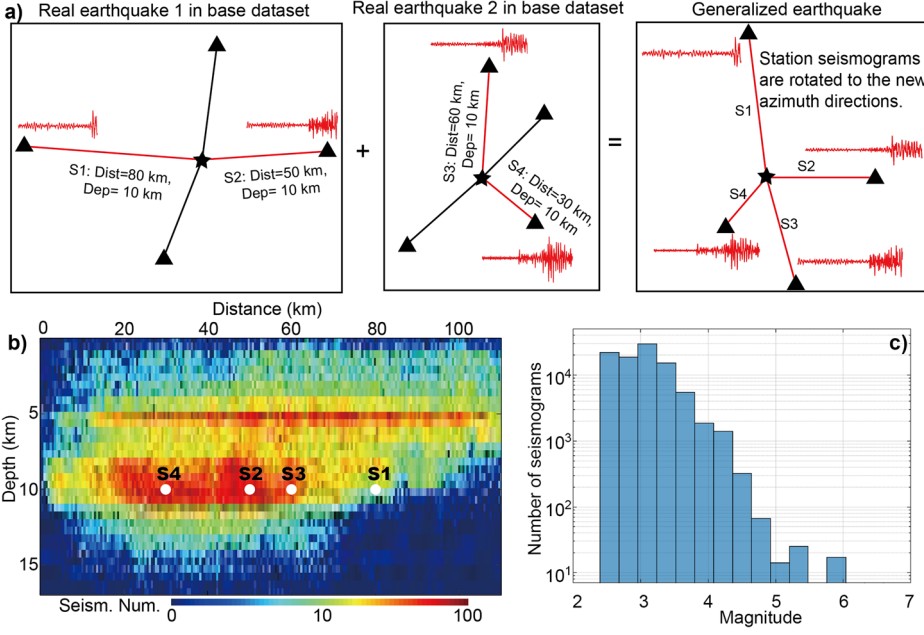

the trained neural network can be applied generally in different areas and monitoring projects.

## Early warning performance for earthquakes in Osaka (Japan) and Ridgecrest (US)

To comprehensively test the performance of the neural networks, we utilize the 179 events in Osaka, Japan, and 349 events in Ridgecrest, US to simulate the early warning progress and assess the errors of the predicted parameters at different times. We truncate the continuous waveforms of the events to different time windows with various lengths of effective signals and utilize the neural networks to predict the corresponding earthquake parameters (Supplementary Fig. 3). Figures 6 and 7 show comparisons between the predicted and cataloged earthquake parameters at the 4th and 15th seconds after the first triggered stations. In the comparison, we do not consider the PDF threshold values and only analyze the magnitude and location results predicted at the specified moments. At the 4th second, the epicentral distance and magnitude errors are 4.8 km and 0.18 for the Osaka earthquakes, and 6.3 km and 0.20 for the Ridgecrest earthquakes, respectively. By the 15th second, the epicentral distance and magnitude errors improve to 2.7 km and 0.05 for the Osaka earthquakes, and 4.9 km and 0.07 for the Ridgecrest earthquakes, respectively. More error statistics at different times show that the earthquake early warning can be activated as early as 3~4 s after the first triggered station, and the accuracy of earthquake parameters generally improves when the monitoring stations receive more data (Supplementary Fig. 3). However, depth resolutions are relatively lower with the monitoring stations at the surface. The depth errors are 2.7 km and 5.0 km for Osaka earthquakes, and 4.1 km and 3.1 km for Ridgecrest earthquakes at 4th and 15th seconds, respectively. When the effective signals are increased to 25 s, the depth errors in Japanese earthquakes are perturbed, as shown in Supplementary Fig. 3. This implies that the complex scattered waves are challenging to learn, as training earthquakes are generalized using simple velocity assumptions while the recombined waveforms originate from various regions. The complex latter waveforms contain abundant features unique to their local regions.

We utilize data from 349 Ridgecrest events to test the traditional picking-based method and compare the results with our neural network models. To closely simulate EEW practices in ElarmS[5–7], we adopt similar parameter settings for phase picking, location, magnitude estimation, and alert criteria whenever possible (Supplementary Table 1). While actual EEW operations in ElarmS are more complex, involving data transmission, event monitoring, and information dissemination, we only implement the data

processing steps from truncated waveforms to earthquake parameters. The data processing workflow reflects the challenges posed by various parameter settings compared to our neural network methods. Supplementary Fig. 4 shows that the mean first alarm time of the neural networks is 0.7 s earlier than that of the traditional method. Additionally, the neural network models yield lower errors in earthquake location and magnitude at the first alarm time. The traditional picking-based method also requires numerous parameter settings (Supplementary Table 1), potentially challenging EEW efficiency and earthquake parameter accuracy compared to neural network models.

## Generalization ability of the neural networks

We test the generalization ability of the neural networks by applying them to different regions across Japan and North California, US (see Fig. 8). One-hour continuous data (starting 90 s before the earthquake occurs) of 130 relatively large onshore earthquakes, monitored by varying numbers of stations, are input into the neural networks. This includes 57 earthquakes from Japan's Hi-net and 73 earthquakes from North California, US, respectively. To test the performance of our application for offshore events with complex velocity structures, we also downloaded the event waveforms of nine earthquakes from Japan's S-net. We set the monitoring areas according to the downloaded station distributions, and ensure that the monitoring areas cover all the stations. We position the center of the monitoring area around the mean values of station coordinates to include as many stations as possible. If the number of stations exceeds 12, we use the closest 12 stations to the epicenter to construct the monitoring area. In the case of Japan earthquakes, most are monitored by 11–12 stations, as shown in Fig. 8, while in Northern California, US, most earthquakes are monitored by 5–8 stations. Figure 8c–e show the error statistics of the predicted epicentral distance, depth, and magnitude for all the earthquakes. The mean errors of the epicentral location, depth, and magnitude are 4.9 km, 4.0 km, and 0.17 for the 130 onshore earthquakes, respectively (Fig. 8). The depth errors are relatively large in the range of 0–5 km possibly due to the weaker constraints imposed by surface stations, similar to traditional methods. However, the mean errors of the Japan offshore earthquakes are 7.3 km, 8.7 km, and 0.22 for epicentral location, depth, and magnitude respectively (Fig. 8), which are larger than the onshore earthquakes and which means that the complex velocity structures would affect the earthquake locations (Supplementary Fig. 5). We detect 4385 earthquakes from all of the one-hour continuous datasets from Japan's Hi-net and North California, US. Although we focus on the EEW of relatively large earthquakes, the results

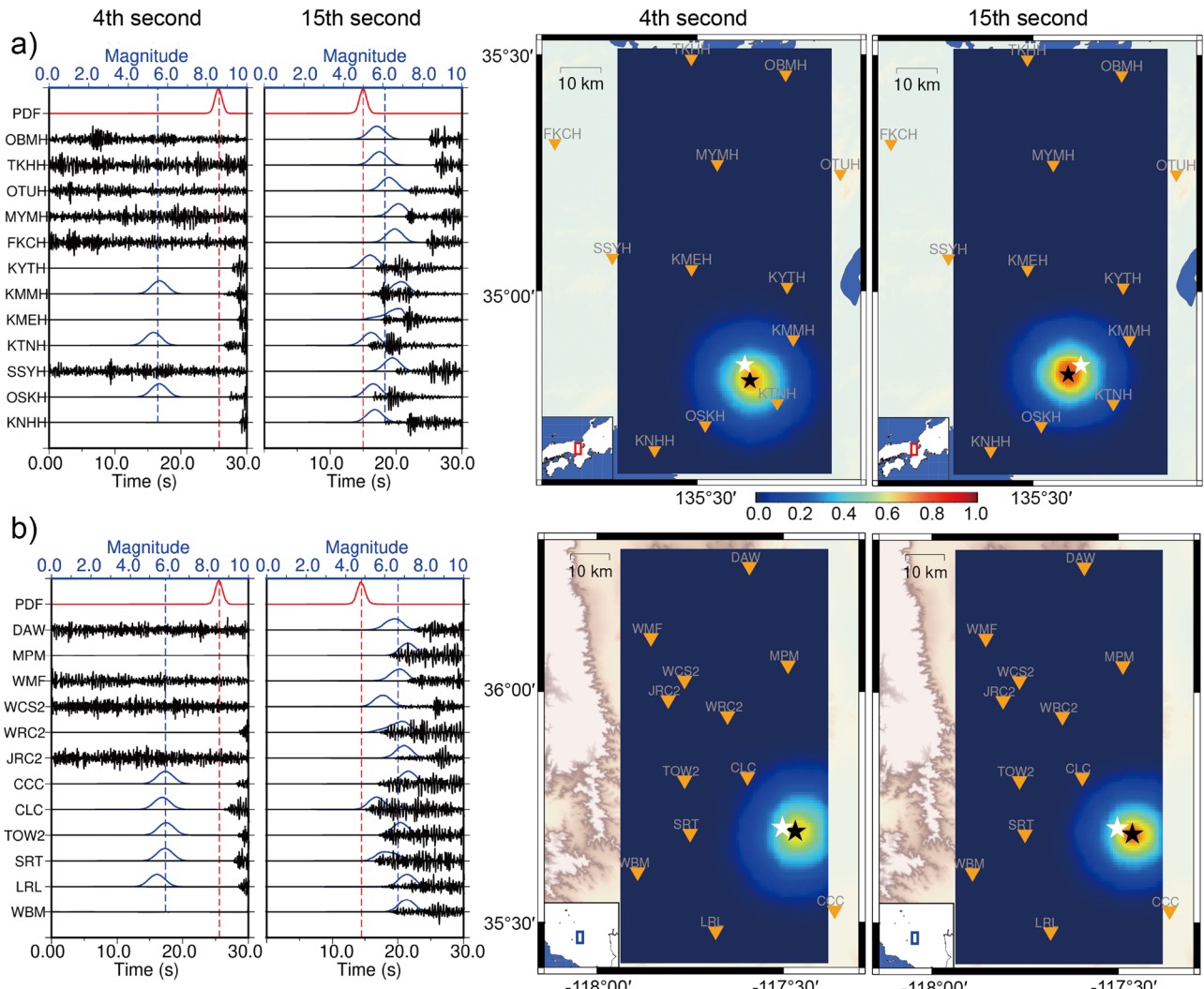

**Fig. 4 | Real-time monitoring of two main shocks. a** The earthquake (M 6.1) occurred on June 18, 2018, in Osaka, Japan. **b** The earthquake (M 6.4) occurred on July 4, 2019, in Ridgecrest, US. The monitoring snapshots at the 4th and 15th seconds after the P arrival of the first triggered stations are shown in the figures. The black, blue, and red curves represent the input waveforms (Z components), magnitude PDFs, and detection PDFs, respectively; the blue dashed lines mark the mean of the predicted magnitudes; the black and white stars in the right figures are the predicted and cataloged earthquake locations; The triangles are the monitoring stations.

reveal that our neural networks can handle smaller earthquakes than those in the training dataset (M ≥ 2.5) (Fig. 8f).

### The first alarms in real-time monitoring with different station distributions

We analyze the monitoring results for 139 relatively large earthquakes, focusing on the first alarms where both detection and location PDF values satisfied threshold criteria of 0.7 and 0.6. Figure 9 illustrates that most first alarms could be issued with approximately 4 s after the first triggered P arrivals. The first alarm time ($T_f$) after the first triggered P arrivals varies based on station number and distributions. For earthquakes monitored by dense station distributions, $T_f$ is relatively small (~4 s), while for those monitored by sparse station distributions (3–5 stations), $T_f$ may exceed 8 s. Despite the varying $T_f$, our results indicate that at the first alarms, the accuracy of earthquake parameters is sufficient for early warning (Fig. 9). The mean errors for epicentral location, depth, and magnitude are 5.5 km, 4.2 km, and 0.32, respectively. The errors in estimated earthquake parameters are similar across cases with different station numbers. Neural networks may require longer effective signals for cases with sparse station distributions. In contrast, dense distributions offer more triggered stations, allowing for better parameter constraints at the earthquake's onset, resulting

in earlier first alarms. The results suggest that neural networks can effectively extract features from various numbers of stations and report alarms instantly when currently available waveforms meet the minimum requirement for constraining the earthquake parameters.

### Application to an M > 7.0 earthquake and its comparison with traditional EEW practice

We apply the trained neural network models to the M 7.3 Kumamoto earthquake in Japan on April 16, 2016. The waveforms are recorded by K-NET and converted to velocity to serve as inputs for the neural networks. Our results show that earthquake parameters are first reported at 3.5 s from the onset of first P arrival, with epicentral location and magnitude errors of 6.1 km and 1.4, respectively (Fig. 10a). Given that the rupture duration for large earthquakes typically exceeds 6 s, waveforms within 3.5 s may not be sufficient to determine the final earthquake magnitude. However, the initial magnitude result of M 5.4 could potentially indicate the arrival of a large earthquake, as the results are reported based on a small amount of signal data. Additionally, although there are artifacts due to truncating the downloaded incomplete waveforms in Fig. 10a, our method robustly determines location and magnitude, as the M 7.3 earthquake's energy dominates the time window. By the 15th second, all stations are triggered,

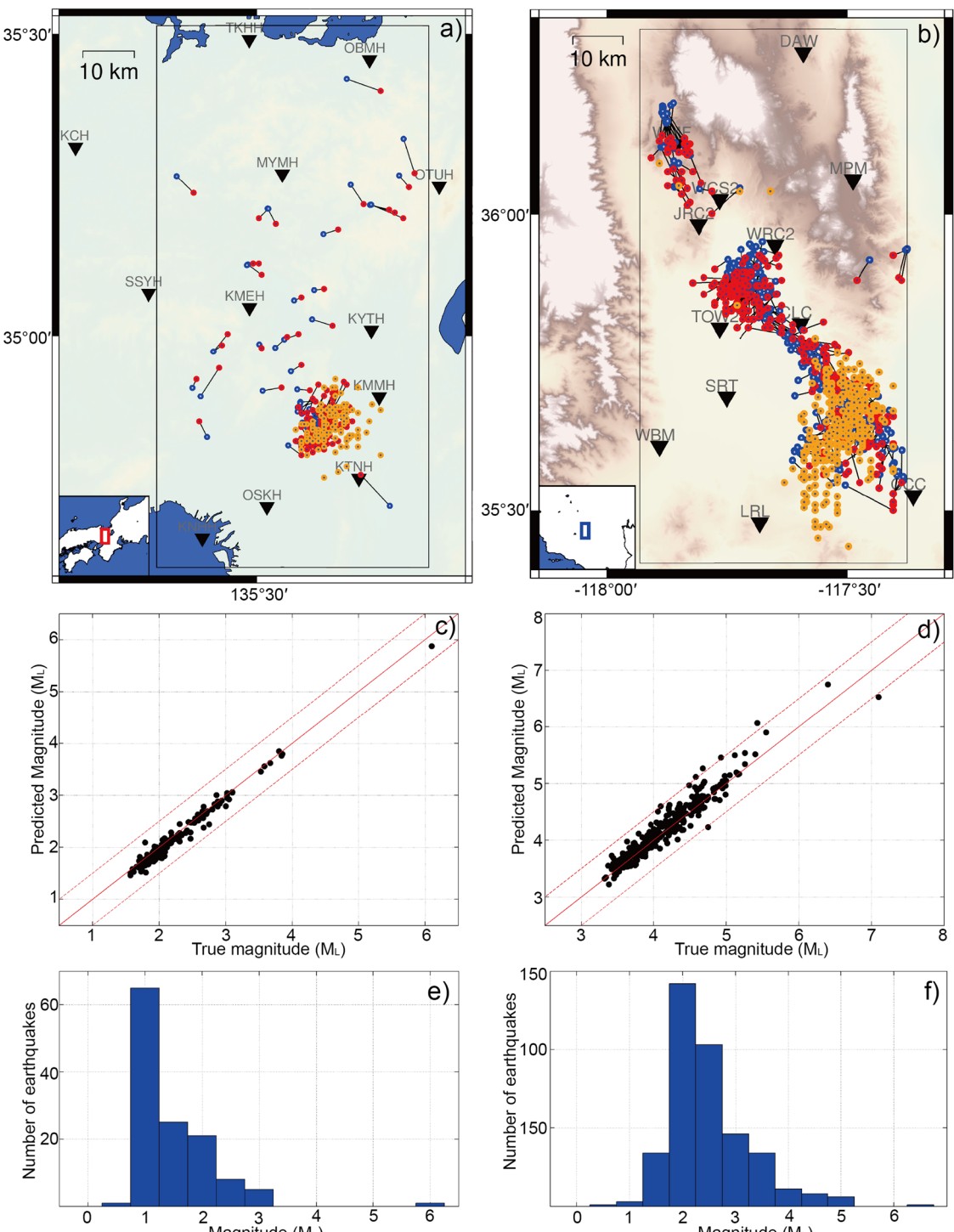

**Fig. 5 | The monitoring results in Osaka, Japan and Ridgecrest, US. a** The predicted results of 179 cataloged earthquakes between January 16 and December 25, 2018, and 126 detected earthquakes (orange) from one-day continuous data on June 18, 2018, in Osaka, Japan. The predicted (red) and cataloged (blue) earthquake locations are connected for comparison. **b** The predicted results of 349 cataloged earthquakes between July 4, 2019, and November 16, 2020, and 389 detected earthquakes (orange) from one-day continuous data on July 4, 2019, in Ridgecrest, US. The predicted (red) and cataloged (blue) earthquake locations are connected for comparison. **c** The magnitude comparison between the cataloged and predicted results of the 179 earthquakes in Osaka, Japan. **d** The magnitude comparison between the cataloged and predicted results of the 349 earthquakes in Ridgecrest, US. **e** The magnitude statistics of the one-day monitoring in Osaka, Japan. **f** The magnitude statistics of the one-day monitoring in Ridgecrest, US.

resulting in location and magnitude errors of 5.0 km and M 7.1, respectively (Fig. 10a). Despite most training data being velocity from broadband seismometers, our models performed well with strong motion seismometers. We also compare the results with the traditional EEW practice of the same earthquake documented in Kodera et al.[52]. Besides the K-NET

stations, the Hi-net stations may also be applied to help determine the hypocenter in the traditional method. Since the time required to obtain a precise hypocenter may be relatively long for the traditional method, a warning is issued based on predefined area divisions consisting of 188 areas across Japan. Although the definitions of time efficiency are different—we

**Fig. 6 | The earthquake parameter comparisons between the predicted results and the cataloged results of 179 earthquakes in Osaka, Japan. a** The earthquake location comparison at the 4th second after the first triggered station. **b** The earthquake location comparison at the 15th second after the first triggered station. The predicted locations (red) are connected with the cataloged locations (blue), while the gray points represent predicted locations with errors larger than 15 km due to low PDFs for current time windows. **c** The earthquake magnitude comparison at the 4th second after the first triggered station. **d** The earthquake magnitude comparison at the 15th second after the first triggered station.

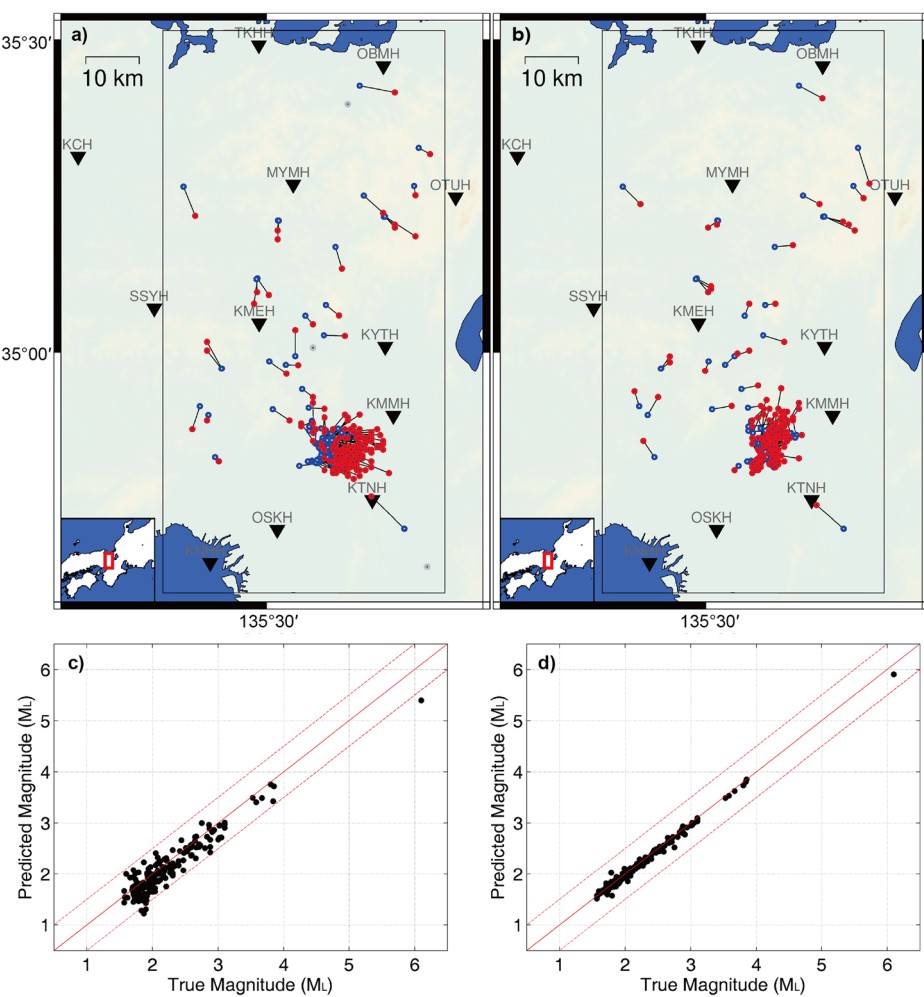

measure the efficiency from the onset of the first P arrival, while Kodera et al.[52] measure the lapse time after the detection operation of the first triggered station—we can compare the absolute issuance time. In the traditional method, the first issuance time is 01:25:14 with a magnitude of M 5.9, which is 2 s later than the neural network model to report the same magnitude as shown in Fig. 10b. The magnitude is updated to M 6.9 at 01:25:18.7 for the traditional method; however the final magnitude approaches the true magnitude at 01:25:16.9 for the neural network model, which is also approximately 2 s earlier than tradition method (Fig. 10b). We calculate the mean magnitude from the triggered stations as the final result, and the early magnitude estimation may be even better by weighted averaging the magnitudes in consideration of different signal lengths in different triggered stations.

## Discussion
### Potential for global real-time EEW application
We trained the neural networks using data from Italy, Oklahoma, and Southern California. They can now be applied for real-time earthquake early warning in different regions without additional training, offering broad applicability to reduce seismic hazards globally. The training data from Southern California were generated before occurrences of the testing earthquakes in Ridgecrest. Despite the temporal difference between the training and testing earthquakes, the geological structures and waveforms may exhibit similarity, given that Ridgecrest is geologically part of Southern California. Nevertheless, the monitoring errors for Ridgecrest earthquakes are similar to the results observed for earthquakes in Japan and Northern California which have no spatio-temporal overlap. This suggests that the neural networks have learned generalized features for earthquake detection

and parameter evaluation, rather than specific features of a particular local area. These models offer a straightforward way to monitor earthquakes in any region without complex empirical settings. Although threshold values for Gaussian distribution PDFs need to be set, the same values can be used across regions without considering station distributions and monitoring settings. We determine the threshold values for Gaussian distribution PDFs by evaluating the predicted labels of various test samples. Our findings indicate that the current settings can ensure robust event detection and accurate earthquake parameter solutions. In comparison with traditional travel time-based methods, which often involve lengthy workflows requiring numerous parameter settings, the efficiency and precision of these methods depend on carefully chosen threshold values and settings tailored for different situations. The generalization ability facilitates the deployment of earthquake early warning systems. Moreover, the neural networks can mine the real-time waveform streams to the utmost and report alarms automatically.

### Limitations and potential improvements
We make use of the move-out features represented by the P arrival times from sorted stations and utilize the convolutional neural networks to extract the earthquake location features from both waveforms and station coordinates. We avoid extracting features separately for each station, as is done in the graph neural network[42,46]. This decision is based on the consideration that the images represented by the sorted waveforms exhibit similarity for earthquakes with similar locations, implying a strong constraint on earthquake locations. The convolutional layer is indeed a powerful tool for extracting move-out features between stations. However, the neural networks could potentially benefit from various structures. Integrating

**Fig. 7 | The earthquake parameter comparisons between the predicted results and the cataloged results of 349 earthquakes in Ridgecrest, US. a** The earthquake location comparison at the 4th second after the first triggered station. **b** The earthquake location comparison at the 15th second after the first triggered station. The predicted locations (red) are connected with the cataloged locations (blue), while the gray points represent predicted locations with errors larger than 15 km due to low PDFs for current time windows. **c** The earthquake magnitude comparison at the 4th second after the first triggered station. **d** The earthquake magnitude comparison at the 15th second after the first triggered station.

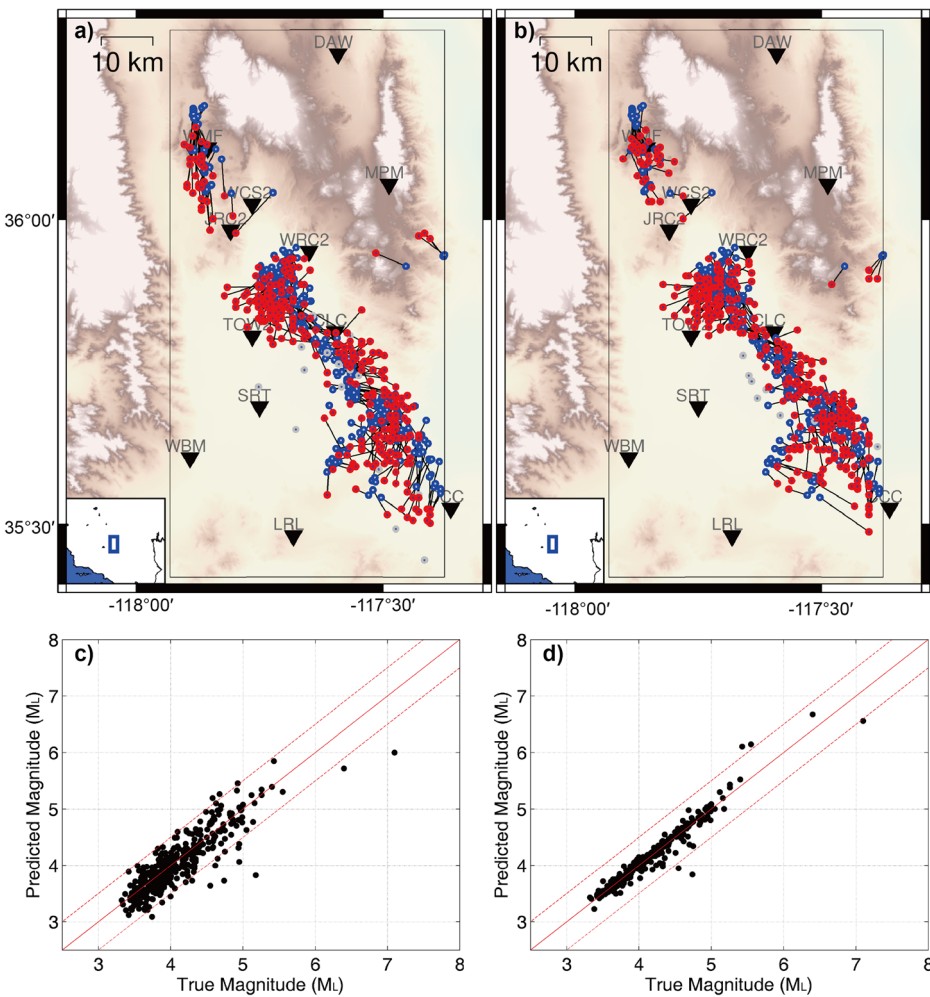

convolutional layers for extracting local features and transformer structures for capturing overall features seems like a promising approach[53]. This combination could potentially enhance the neural network's ability to learn both specific details and global patterns, providing a more comprehensive understanding of earthquake features.

The monitoring system may not be able to distinguish multiple events that occurred in the same input window of 30 s (Supplementary Fig. 2). The neural networks may output the results of the largest earthquake with dominating signals in the input time window, potentially missing the small earthquakes. In another aspect, if multiple events with similar magnitude are mixed in the same window, the outputted PDFs may be very small and the earthquakes could not be distinguished, or the output earthquake parameters from different events are mixed, which may result in false alarms of the earthquakes. This poses a challenge in earthquake monitoring for small earthquakes, even though large earthquakes rarely occur at the same time in a monitoring area. A potential solution to this problem is to stack two earthquake samples together to simulate cases with two earthquakes occurring nearly.

Onsite early warning methods commonly use 3–4 s waveforms from a single station to assess local ground motion or magnitude[54,55]. The teleseismic signal poses a challenge when relying on a single station for EEW. Traditionally, teleseismic signals can be distinguished based on frequency features, as the high-frequency energy is significantly attenuated for teleseismic signals[7]. In our application, we generate 2000 earthquakes outside of the monitoring ranges as abnormal events to train the neural networks. Earthquakes outside of the monitoring areas exhibit very different features from normal events in terms of the waveforms

from multiple stations. Incorporating more teleseismic waveforms for training could further enhance the efficiency of the network models. The neural networks may be further generalized by incorporating other information for training. For instance, smartphones, which cover high-population regions, have been proven to be applicable for EEW, and signals from accelerometers in smartphones could be utilized to detect earthquakes[56].

The current model requires the station distribution to cover the monitoring areas, which may limit its application for monitoring offshore events using onshore stations. The neural network models could be further improved by extending the monitoring area and increasing the diversity of station distributions to accommodate offshore earthquake monitoring. Our generalization method provides a framework to generate training samples in arbitrary monitoring settings, given sufficient single-station waveforms. However, the current assumption is based on a global 1D layered model to recombine the waveforms, which may increase location errors for offshore earthquakes with complex velocity structures (e.g., subduction zones). Possible improvements include applying data recombination using single-station waveforms from a specified area or using exact cataloged events in that area to further fine-tune the generalized models. Additionally, incorporating station elevation as an input channel could help accommodate large topographic variations, and increasing the number of stations could support larger monitoring areas.

The earthquake depth solutions are affected by complex velocity structures and are less sensitive to arrival time features from multiple stations compared to epicentral locations. Although we focus on relatively shallow earthquakes, as they are more destructive, determining the depth of

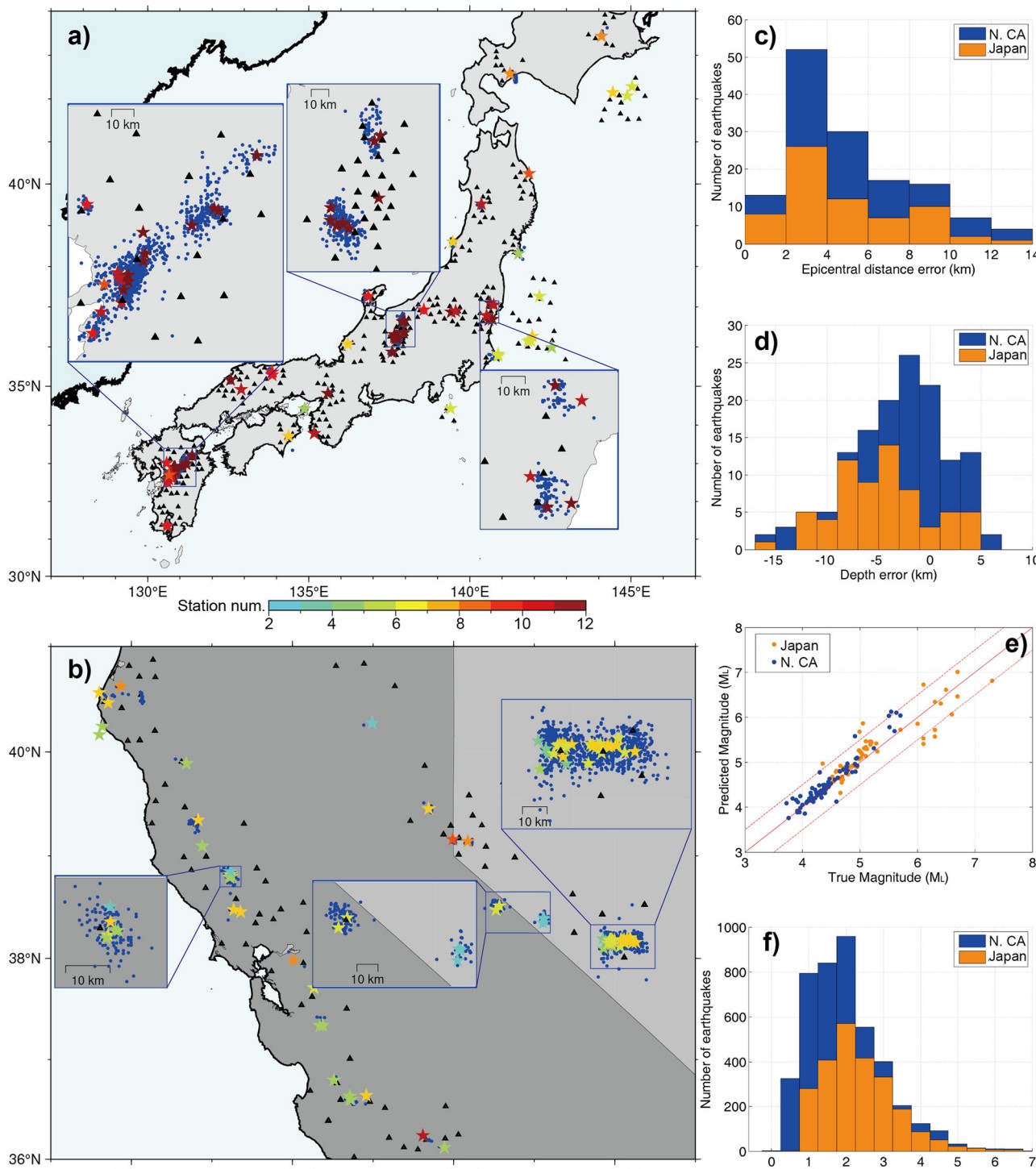

**Fig. 8 | The real-time monitoring of the neural networks to the 139 relatively large earthquakes that occurred in different regions. a** Fifty-seven M ≥ 5.0 onshore earthquakes and nine M ≥ 5.0 offshore earthquakes in Japan. **b** Seventy-three M ≥ 4.0 earthquakes in Northern California, US. The stars are the predicted locations with the color indicating the number of monitoring stations, and the blue dots are the detected earthquakes from one-hour continuous data following the large earthquakes. The black triangles are the selected monitoring stations for the earthquakes. **c** Epicentral distance errors. **d** Depth errors. **e** The comparison between the predicted and true magnitudes. **f** The magnitude statistics of the detected earthquakes for both regions.

deep subduction earthquakes is more challenging. A possible improvement could be the development of an extra neural network specifically for depth determination from single-station waveforms while using multiple-station waveforms to determine the epicentral location. This approach could simplify training and allow the neural network to leverage single-station waveform features more effectively for depth determination.

The earthquake parameter solutions may be further improved by incorporating more data from different regions and types of instruments globally to increase the diversity of the training samples. Since small earthquakes are always more common than large earthquakes in the training set, the magnitude error for large earthquakes may be greater than for relatively small earthquakes[36]. Additionally, large earthquakes are

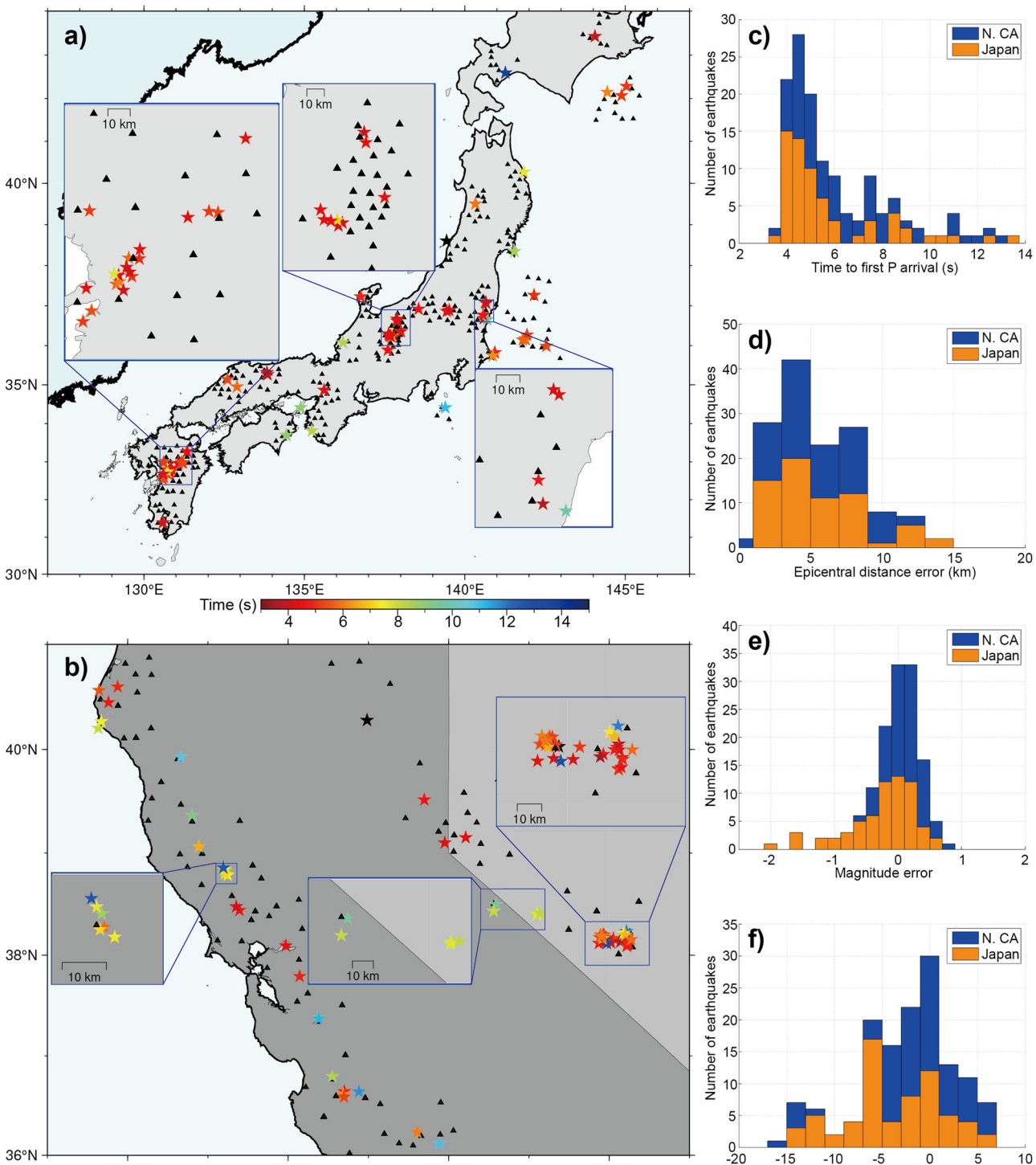

**Fig. 9 | Predicted earthquake parameters at the first alarms for the 139 relatively large earthquakes. a** Time distribution of the first alarms in Japan. **b** Time distribution of the first alarms in N. California, US. **c** The time of the first alarms. The time of the first alarm is defined as the difference between the P arrival time of the first triggered station and the current time. **d** The epicentral error. **e** The magnitude error. **f** The depth error.

mainly recorded by strong-motion accelerometers, while small earthquakes are recorded by broadband seismographs in our datasets. Including more strong-motion data across various magnitudes may further improve magnitude accuracy. In this study, we augmented the large earthquake samples by randomly shifting the truncating time window for magnitude network training; however, increasing the magnitude diversity of the original samples by using data from various areas could further enhance the generalization ability. Our models are tested using

continuous data without considering network transmission, and the data may contain glitches, gaps, and clipping. The abnormal signals (e.g., data gaps, spike signals) in the noise waveforms are easily distinguished by the neural network, and using more such abnormal signals as training samples could effectively avoid false alarms (mistaking noise for earthquake events). However, if abnormal signals appear in the earthquake waveforms, they may affect the accuracy of the earthquake parameters. For example, small earthquakes with large-amplitude spike signals may be

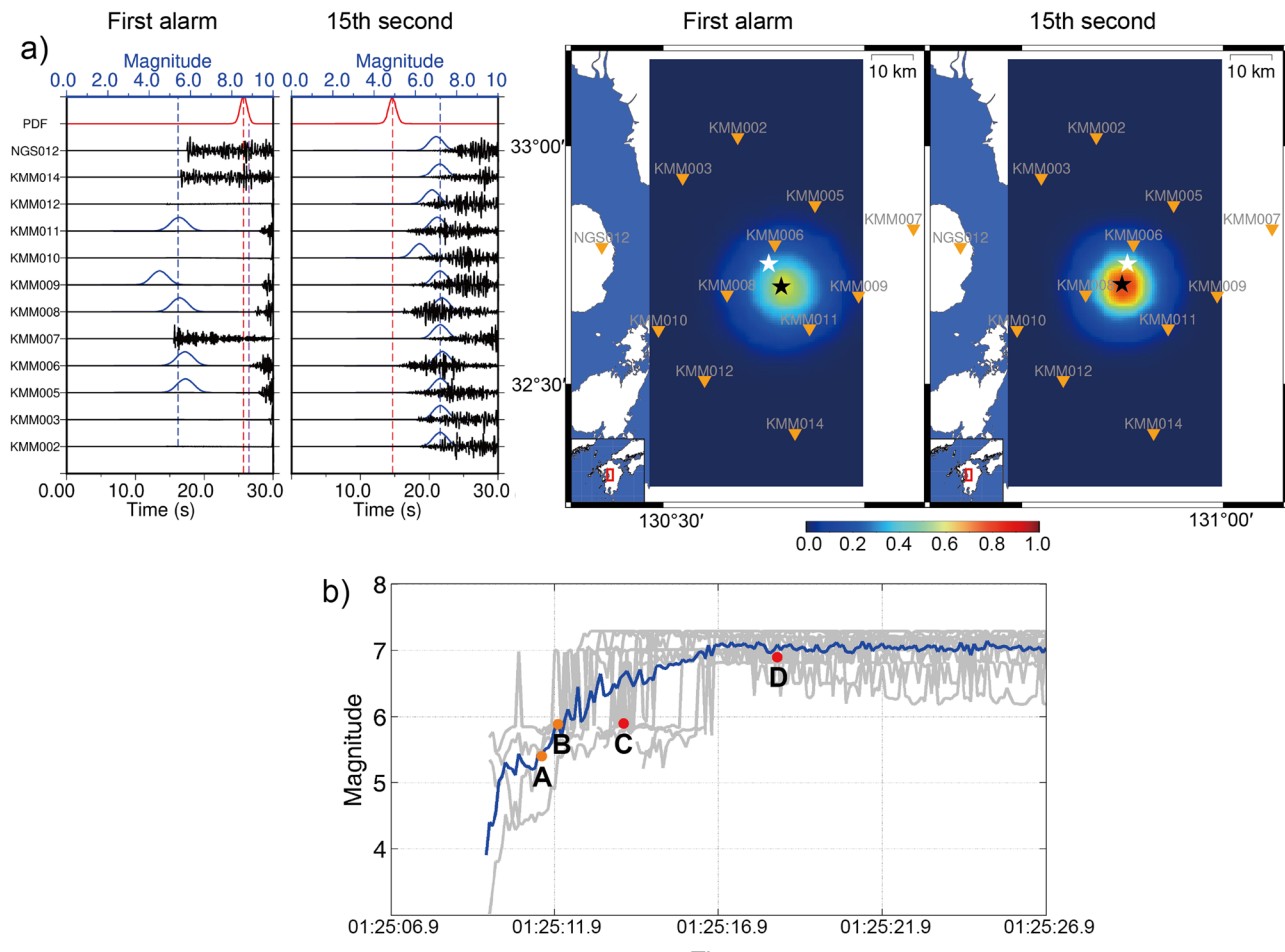

**Fig. 10 | Application to the M 7.3 Kumamoto earthquake in Japan on April 16, 2016. a** The monitoring snapshots at the first alarm and 15th second from the first P arrival. The black, blue, and red curves represent the input waveforms (Z components), magnitude PDFs, and detection PDFs, respectively; the blue dashed lines mark the mean of the predicted magnitudes; the black and white stars in the right figures are the predicted and cataloged earthquake locations. **b** The predicted magnitude at different times. The gray curves represent the predicted magnitudes for 12 monitoring stations, while the blue curve shows the mean magnitude. At points A, the detection and location PDFs are 0.95 and 0.60, and the system reports the first alarm. The magnitudes before point A are the results with location PDF below 0.6. At point B, the detection and location PDFs are 0.96 and 0.68, and the predicted magnitude is M 5.9 for comparison with the traditional method (Kodera et al.[52]). For the traditional method, the initial issuance time is 01:25:14.0 with a magnitude of M 5.9 (Point C). The magnitude is then updated to M 6.9 at 01:25:18.7 (Point D).

predicted as large earthquakes, leading to false alarms. Although our models are applicable for clipping data from broadband seismographs (e.g., M 6.4 Ridgecrest earthquake), performances could be further improved by incorporating more training samples with abnormal signals from real-time waveform streams.

## Data and methods
### Training and testing data background
The training data primarily consists of central Italy earthquakes from January 15, 2010, to September 25, 2018, and Oklahoma-induced earthquakes from April 16, 2013, to March 31, 2016. We collect waveform datasets from 4285 central Italian earthquakes (M ≥ 2.5) recorded by 12 permanent broadband seismographs and from 1265 Oklahoma earthquakes (M ≥ 3.0) recorded by 30 permanent broadband seismographs (Fig. 2). Most of the earthquakes in Italy occur at depths ranging from 5 to 17 km, while the induced earthquakes in Oklahoma occur at comparably shallow depths, ranging from 0 to 8 km. The total number of seismograms for the Italy dataset ideally should be 51,420 (12 × 4285), while the Oklahoma dataset should have 37,950 (30 × 1265) seismograms. However, due to missing data in some stations and events, the final number of seismograms is 42,945 for Italy and 36,203 for Oklahoma. The corresponding epicentral distances of the collected seismograms are

mostly in the range of 0–110 km. To further increase the diversity of the seismograms, we collect data from 768 earthquakes (M ≥ 2.5) that occurred before July 4, 2019, in Southern California (Fig. 2). However, many of the datasets from the stations are missing, and we are only able to collect 15,438 training seismograms recorded by broadband seismographs. Therefore, the total number of collected seismograms is 94,586 (42,945 + 36,203 + 15,438), and the depth and epicentral distance distributions are shown in Fig. 3b. Since the majority of the collected earthquakes are relatively small, we included 21 M > 6.0 large earthquakes that occurred in Japan between October 6, 2000, and April 11, 2011, incorporating 1454 single-station waveforms to aid in the training of the magnitude network. Although the portion of large earthquakes in the training dataset is relatively small, we enhance their significance by data augmentation for the magnitude network training. The large earthquakes are recorded by strong motion seismometers, and we convert the corresponding waveforms from acceleration to velocity. All waveforms are resampled to 20 Hz, and the three components are rotated to the E, N, and Z directions. We remove the mean and linear trend from the waveforms and eliminate the instrument responses by deconvolution. To ensure reliable earthquake detection, location, and magnitude estimation, all waveforms are bandpass-filtered using a range of frequencies optimized for these tasks. Specifically, a filter range of 2–8 Hz is

https://doi.org/10.1038/s43247-024-01718-8 **Article**

used for earthquake detection and location, while a range of 0.5–9 Hz is used for magnitude estimation.

To simulate the early warning process, we truncate the waveforms into different time windows to mimic the scenario when there are only partial earthquake signals from the triggered monitoring stations. We randomly cut the waveform of an earthquake sample starting from 1.0 to 26 s relative to the first triggered P phases among all stations, with a total truncating window of 30 s. We randomly generated 355,001 earthquake samples across the monitoring area with a varying number of monitoring stations distributed within a specific range. We also generate an additional 2000 earthquake samples outside the monitoring area. These are intentionally labeled with a probability density function (PDF) of zero to represent abnormal events, thereby enabling the neural network to distinguish earthquakes occurring outside the monitoring area. These samples are then utilized for training the detection and location networks. For the magnitude network training, we utilize single-station waveforms as input directly. We generate 200,000 samples from the 94,586 base waveforms by randomly cutting the time windows. To emphasize the significance of large earthquakes, we generate 100,000 additional samples from the 1454 base waveforms of M > 6.0 earthquakes using the same augmentation method. This ensures the model is well-trained to accurately assess the magnitude of large earthquakes.

To evaluate the real-time monitoring efficiency and generalization ability of the trained network models, we apply them to the 2018 Osaka $M_{JMA}$ 6.1 earthquake sequence in Japan, the 2019 Ridgecrest earthquake sequence in the US, and 139 relatively large earthquakes across Japan and Northern California. On June 18, 2018, at 07:58 Japan Standard Time, a magnitude 6.1 earthquake struck the Osaka metropolitan area, which has a population exceeding eight million. The earthquake generated strong ground motions and resulted in four casualties. The $M_w$ 6.4 earthquake sequence that occurred on 4 July 2019 also caused damages to at least 100 homes and businesses in the communities of Ridgecrest and Trona. For the Osaka earthquake sequence on June 18, 2018, we downloaded one day's continuous waveform recorded by 12 stations within a range of 80 km to 100 km to test the efficiency of the neural networks. In addition, we selected 179 earthquakes ($M_{JMA} > 2.0$) that occurred between January 16 and December 25, 2018, to assess the accuracy of the predicted earthquake parameters by generalized neural networks. According to the routine catalog provided by the Japan Meteorological Agency (JMA), the aftershocks following the main shock of $M_{JMA}$ 6.1 are relatively weak, with magnitudes not exceeding $M_{JMA}$ 4.1. We primarily focus on the early warning of relatively large earthquakes, although the neural network is also capable of detecting smaller earthquakes. For the Ridgecrest earthquake sequence, we also downloaded continuous seismic data recorded by 12 nearby stations on July 4, 2019, and 349 relatively large earthquakes (M ≥ 2.5) from July 4, 2019, to November 16, 2020, to assess the performance of the neural networks. In the downloaded dataset, some stations are malfunctioning or missing; nevertheless, the neural network is capable of handling datasets with a flexible number of stations (≤12).

To assess the generalization ability, we apply the trained models to 57 relatively large onshore earthquakes ($M_{JMA} \geq 5.0$) in Japan and 73 relatively large onshore earthquakes (M ≥ 4.0) in North California, US. We download one-hour continuous data for each earthquake in both regions, recorded by a varying number of stations. To assess the performance of the neural networks applied to complex velocity models, we downloaded event data from nine offshore earthquakes (M > 5.0) monitored by Japan's S-net (Ocean-Bottom Seismograph). To compare the performance with traditional EEW practices, we downloaded the waveforms of the M 7.3 Kumamoto earthquake in Japan on April 16, 2016, recorded by K-NET (Strong-motion seismograph). For all the earthquakes, we downloaded the waveforms of the stations with epicentral distances less than 110 km. We select the stations without excessive missing data and components, and set the monitoring area for each earthquake separately to ensure that the station range includes as many stations as possible. If the number of stations exceeds 12, we select the closest 12 stations to the epicenter for testing.

## Model architectures and training

We design two neural networks for the detection and location of earthquakes using 2D convolutional layers, as illustrated in Fig. 1. The input for the detection neural network consists of three components of waveform data from multiple stations. The input size (12 × 1024 × 3) is determined by the maximum number of stations, the maximum length of time samples, and the number of components. Specifically, the total length of each waveform is 30 s with a time interval of 0.05 s, resulting in 600 time samples. To meet the requirements of MaxPooling and UpSampling, which need the time samples to be in the form of $2^n$, the remaining 424 (1024-600) time samples are padded with zeroes. In our application, the number of stations can vary (less than 12), and any additional dimensions beyond the station number are also padded with zeroes. The output of the neural network is labeled with a Gaussian distribution, with the peak value corresponding to the arrival time of the first triggered P phase if an earthquake event is detected. If the input contains only noise, the output is labeled with zeroes.

For the location neural network, the input includes both the XY coordinates of the stations and their corresponding waveform data. The station X and Y coordinates are normalized to a range of 0–1 based on the station ranges (0–82 km and 0–100 km). These normalized coordinates are represented as two vectors, each with a length of 1024 samples, occupying two channels along with the corresponding waveform data. In contrast to the graph neural network approach[33,37], we utilize the move-out features of the waveforms and organize the input data by sorting stations based on their X and Y coordinates in ascending order. Consequently, the total size of the input is 12 × 1024 × 10, with the first 5 channels containing the three waveform components and station locations sorted by X coordinates, while the remaining 5 channels contain the same data sorted by Y coordinates. This arrangement results in that the input image pattern is unique for an earthquake monitored by the specified stations. The output of the location neural network is labeled with a 3D Gaussian distribution, with the peak value representing the predicted location. The grid size corresponds to the monitoring range of 16–66 km for X, 0–100 km for Y, and −6–22.8 km for depth. The training earthquakes are generated with depths ranging from 0 to 20 km. However, for labeling the output, we employ a broader depth range, taking into account that the Gaussian distribution radius at the boundary of deep or shallow earthquakes may exceed the depth range if the same range is utilized for labeling.

Regarding the magnitude network, we calculate the magnitude for each station individually, with an input size of 1024 × 5. The first three channels consist of normalized waveform data, and the normalization factor is the maximum amplitude in m/s. The fourth channel represents the epicentral distance and the fifth channel is the logarithm of the normalization factor for the waveform. Similar to the input for the detection and location neural networks, the 30-s seismic waveforms occupy 600 time samples, and any remaining 424 time samples are padded with zeros. Additionally, the epicentral distance is normalized to a range of 0–1, corresponding to distances from 0 to 110 km. Similarly, the logarithm of the normalization factor is also scaled to a range of 0–1, corresponding to values from −10.0 to 0.0. We randomly truncate the waveforms starting from 1 to 25 s relative to the P arrival time for network training, ensuring that the final model can predict the magnitude with only a few earthquake signals within the input time window. The output of the magnitude model is labeled with a 1D Gaussian distribution, where the peak value marks the predicted magnitude.

The neural network architectures predominantly utilize Convolutional, MaxPooling, and Upsampling layers, as illustrated in Fig. 1. The design of the detection and location neural networks incorporates 2D layers to handle input data from multiple stations. In contrast, the magnitude network consists of 1D convolutional layers to process data from individual stations. The kernel sizes for the 2D and 1D convolutional layers are uniformly set to 3 × 3 and 3, respectively. Zero-padding is applied to the output of each convolutional layer to maintain the output size.

MaxPooling layers are employed to extract critical features for constraining earthquake parameters, while Upsampling layers are used to adjust the final output size of the network model. The task of earthquake detection is relatively straightforward compared to the location and magnitude estimation. Thus, a fully convolutional neural network is sufficient to achieve the goal of earthquake detection. To mitigate the issue of gradient vanishing, we introduce multiple copy layers into the location and magnitude neural networks, as shown in Fig. 1.

The three models are trained using the Adam algorithm with a learning rate of $10^{-4}$. Additionally, they are equipped with 2, 4, and 2 Dropout layers in the detection, location, and magnitude networks, respectively[57]. The trained detection and location network models are merged into a single file, enabling simultaneous detection and location. Although parallel running of the detection and location networks may require more computational resources for all the real-time windows, running them in series, with localization waiting for detection, can result in increased processing time. When applying the models to a monitoring region, we first set the origin of coordinates to ensure that the monitoring stations adequately cover the monitoring area. We then convert the geographical coordinates to relative Cartesian XY coordinates (km). The neural network outputs the earthquake locations in Cartesian XY coordinates, which we subsequently convert back to the corresponding geographical coordinates. When the merged neural network detects an earthquake event, it calculates the theoretical P arrival times. The triggered stations are determined based on whether the P arrival times fall within the monitoring time window. The waveform data from the triggered stations are then passed to the magnitude network to estimate magnitudes, and the final magnitude result is the mean value across all triggered stations.

## Data availability

Seismic data from Italy and the USA were downloaded from Italy's National Institute of Geophysics and Volcanology (INGV), Northern California Earthquake Data Center (NCEDC), Southern California Earthquake Data Center (SCEDC), IRIS Data Management Center (IRIS) through the International Federation of Digital Seismograph Networks (FDSN) web services (https://www.fdsn.org/datacenters/). Seismic data from Japan were downloaded from Hi-net (https://hinetwww11.bosai.go.jp/auth/?LANG=en) and K-NET (https://doi.org/10.17598/NIED.0004) operated by the National Research Institute for Earth Science and Disaster Resilience (NIED). The earthquake catalog used in this study can be downloaded from the following links: http://cnt.rm.ingv.it/ (earthquakes in Italy), http://earthquake.usgs.gov/earthquakes/search/ (earthquakes in the USA), https://www.data.jma.go.jp/svd/eqev/data/bulletin/index_e.html (earthquakes in Japan).

## Code availability

The generalized neural networks, codes, and demo examples can be accessed via the following link: https://github.com/zxiong218/EEWNet. The package Obspy is utilized to process the seismic data (https://docs.obspy.org/).

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

## Acknowledgements

This work was supported by National Natural Science Foundation of China (Grant Nos. 42474092 and U2239204 to X.Z.), Jiangxi Provincial Natural Science Foundation (Nos. 20224BAB211024 and 20242BAB25190 to XZ), Shanghai Sheshan National Geophysical Observatory (No. SSOP202103 to X.Z.), and the Natural Sciences and Engineering Research Council of Canada Discovery Grant (RGPIN-2019-04297 to M.Z.).

## Author contributions

X.Z. designed the project, developed the method, processed the data, and analyzed the results. M.Z. analyzed the results. X.Z. wrote the manuscript. M.Z. revised the manuscript.

## Competing interests

The authors declare no competing interests.
