## [Peer Review file · Communications Earth & Environment]

Universal neural networks for real-time earthquake early warning trained with generalized earthquakes

Corresponding Author: Professor Xiong Zhang

Attachments originally supplied by the reviewers can be found at the end of this file, in order by reviewer number and revision

Version 0:

Decision Letter:

Dear Professor Zhang,

Your manuscript titled "Universal Neural Networks for Real-Time Earthquake Early Warning Trained with Generalized Earthquakes" has now been seen by 3 reviewers, whose comments are appended below. You will see that they find your work of some potential interest. However, they have raised quite substantial concerns that must be addressed. In light of these comments, we cannot accept the manuscript for publication, but would be interested in considering a revised version that fully addresses these serious concerns. In particular, please ensure that the revised manuscript meets the following editorial thresholds:

**** Benchmark the proposed EEW system against current operational practices for efficiency and precision, providing a detailed comparison to highlight the advantages and limitations of your approach. ****

**** Address the scalability and generalizability of your model to larger earthquakes ($M > 6$) and different geological settings, including deep subduction zone earthquakes, with specific examples and data to support your claims. ****

**** Thoroughly discuss the trade-offs between speed and accuracy in EEW systems, especially concerning the detection and magnitude estimation of large earthquakes. Include a detailed examination of how your model manages these challenges and is a clear improvement upon existing approaches. ****

We hope you will find the reviewers' comments useful as you decide how to proceed. Should additional work allow you to address these criticisms, we would be happy to look at a substantially revised manuscript. If you choose to take up this option, please either highlight all changes in the manuscript text file, or provide a list of the changes to the manuscript with your responses to the reviewers.

If the revision process takes significantly longer than three months, we will be happy to reconsider your paper at a later date, as long as nothing similar has been accepted for publication at Communications Earth & Environment or published elsewhere in the meantime.

Please use the following link to submit your revised manuscript, point-by-point response to the reviewers' comments with a list of your changes to the manuscript text (which should be in a separate document to any cover letter), a tracked-changes version of the manuscript (as a PDF file) and any completed checklist:

Link Redacted

**** This url links to your confidential home page and associated information about manuscripts you may have submitted or be**

reviewing for us. If you wish to forward this email to co-authors, please delete the link to your homepage first **

Please do not hesitate to contact us if you have any questions or would like to discuss the required revisions further. Thank you for the opportunity to review your work.

Best regards,

Luca Dal Zilio
Editorial Board Member
Communications Earth & Environment

Joe Aslin
Deputy Editor
Communications Earth & Environment

EDITORIAL POLICIES AND FORMAT

If you decide to resubmit your paper, please ensure that your manuscript complies with our editorial policies and complete and upload the checklist below as a Related Manuscript file type with the revised article:

Editorial Policy Policy requirements
(Download the link to your computer as a PDF.)

For your information, you can find some guidance regarding format requirements summarized on the following checklist: (<https://www.nature.com/documents/commsj-phys-style-formatting-checklist-article.pdf>) and formatting guide (<https://www.nature.com/documents/commsj-phys-style-formatting-guide-accept.pdf>).

REVIEWER COMMENTS:

Reviewer #1 (Remarks to the Author):

The manuscript entitled "Universal Neural Networks for Real-Time Earthquake Early Warning Trained with Generalized Earthquakes" presents a machine learning based approach for earthquake early warning. Its primary innovation lies in devising a new framework to generate model training datasets from real earthquake recordings, ensuring that the resulting model is independent of monitoring geometries. Consequently, the trained model can be applied to earthquake monitoring arrays of different shapes and sensor quantities, thus enabling easy adaptation to various regions. The manuscript is well-written and includes various application cases. My major concerns of this manuscript are that: firstly, the proposed method should be benchmarked with existing EEW practices; and secondly, the limitations of the proposed method should be explicitly addressed in the manuscript. I listed all my concerns below:

1. To highlight the effectiveness and benefits of the proposed approach, authors should benchmark the presented approach with the current operational EEW system or best-practice. For EEW, robustness and efficiency are critical. I noticed the processing time of the proposed approach is not explicitly mentioned. The authors should address how efficiently their method can process the streaming seismic data and generate EEW-related results. For benchmarking, it is important to compare in terms of efficiency (system running time) and precision, and clearly document the parameters and/or criteria used in both methods. In this way, people can better see the advantages of your method. For example, as mentioned in the manuscript, the authors claim their method depends less on parameters and criteria selection.

2. The limitations of the proposed method are not summarized and discussed explicitly in the manuscript. For people to better understand and use the proposed approach, it is necessary to address the limitations somewhere in the manuscript. From my point of view (correct me if I am wrong), some limitations I am aware can be:

(1) this approach adopts a global 1D velocity model. Therefore, location bias may exist for specific regions of distinct velocity models compared to the assumptions used in the training process.

(2) there seems a limitation on the size of the applied region and/or an epicenter distance limitation (as described in Lines 479, and 489).

(3) events should be located inside the monitoring array. If this is true, then this approach does not apply to large offshore events. If this is not true, it would be nice to demonstrate the proposed approach to a mega-thrust event that occurred around Japan's coast.

(4) station elevation seems not encoded into the network, thus large topography is not considered.

(5) there is a maximum number of stations that can be adopted.

3. The authors do not mention the model is trained on broadband sensors or strong motion sensors. From my perception, it is

likely broadband sensors. So how well can the trained model be generated to data recorded by strong motion sensors? For large events, strong motion sensors will not be saturated and can record signals in high quality.

4. In Figure 2, S2 and S3 do not correspond in the subfigure A and B.

5. In lines 157-158, the thresholds of detection and location are set to be 0.6 and 0.7 respectively. How these parameters are determined and are they universally applicable in various applicable scenarios?

6. Lines 123-125: it seems the event will always be located in X: 16-66 km and Y: 0-100 km. Does this mean the system will only monitor events inside this region? What if for a monitoring system, events may occur in arbitrary locations, for example in region X: 1-10 km or 80-100 km?

7. When generating training samples, the authors compile generalized earthquakes by the combination of event waveforms from different stations and earthquakes (as shown in Figure 2). During this process, why do we need to rotate the original stations to a new azimuth? What are the criterians to perform rotations and how to determine the new azimuth angle?

8. When compiling generalized earthquakes, do you use and combine stations from the same region (in your case, Italy, SC, and Oklahoma), or do you mix the stations from different regions? If you combine event waveforms from different regions, the velocity model difference may contribute to distinct P moveouts which may smear your location problem.

9. Fig S3 shows the epicenter and depth errors of Okasa keep increasing after ~5 seconds of receiving data. But in comparisons, the Ridgecrest case shows these errors keep decreasing until ~15 seconds of receiving data, and then gradually increase afterward. The authors attribute this contradictory phenomenon to the effect of complex scattered waves generated by geological structures. Is this explanation supported by any geological or geophysics evidence? Please discuss this in more detail to support this argument.

10. Fig S4, it is better to describe the magnitudes of the two earthquakes in the figure caption.

11. For the detection network, it is described: "The input size ($12 \times 1024 \times 3$) is determined by the maximum number of stations, the maximum length of time samples, and the number of components. Specifically, the total length of the waveform is 30 seconds, with a time interval of 0.05 seconds, resulting in 600 time samples. Any remaining 424 time samples are padded with zeroes." I am wondering why not use 600 samples directly but rather use 1024 with the remaining 424 zeros. Process less but meaningful data would be more efficient and sensible.

Reviewer #2 (Remarks to the Author):

Dear authors and editor, thanks for an interesting article. The paper presents an earthquake early warning algorithm that is scalable to different geographic scenarios using data recombinations and deep learning neural networks. My concern is, the earthquake early warning algorithm is more important for large earthquakes ($M > 6$), which are not included in the training database. I have some questions and comments as shown below and in the attached PDF (please open with Adobe Acrobat or Microsoft Edge) and hope that these be useful to improve the article.

1. In Fig. 4, earthquakes of M6.1 and M6.4 reveal that the algorithm underestimates magnitudes, and the estimates do not exceed M6 for either example. This discrepancy may stem from sample saturation or the algorithm's inability to extrapolate magnitudes beyond the training range. Could you specify the type of signal recorded by the stations? Are they using broadband velocimeters or strong-motion accelerometers? More important, examining examples with unsaturated signals, such as 2011 M9 Tohoku earthquake or the 2003 M8.3 Tokachi-Oki earthquake, using public data from the K-NET and KiK-net databases, may aid in interpreting the results.

2. If the algorithm is employed in earthquake early warning scenarios, such as in South America where earthquakes in the subduction zones of Chile, Peru, or Ecuador often exceed depths of 40 km, its efficacy may be compromised. This is attributed to the shallow depth range covered in its training, prompting concerns regarding its adaptability and effectiveness in accurately estimating earthquakes at such depths.

3. Considering alert messages are disseminated after station records surpass specific acceleration or magnitude thresholds, reevaluating the time of the first alarm (T_f) would be more relevant if based on when the algorithm estimates a magnitude threshold (e.g., M5 or M6 for alert issuance). This approach aligns with Fig. 9, where all earthquakes are assumed to trigger alert messages. Moreover, exploring T_f for events with magnitudes larger than M6 or M7 becomes crucial for EEWS, because the Source Time Function duration is typically more than 6 seconds (Meier et al. 2017, Science).

Reviewer #3 (Remarks to the Author):

In this paper, the authors developed a deep learning model that can determine earthquake parameters from continuous seismic waveform in real-time. The model is generalized so that it can be applied to different regions with different network configurations and geological settings. They then demonstrate its performance in Japan and California, with location errors of 2.6-6.3 km and magnitude errors of 0.05-0.17, showing the potential of EEW application. The merit of this paper is purely engineering with the goal to shorten the time in data preprocessing, transfer learning (re-train in a different geological region), and minimal subjective decisions in real-time source parameter determinism.

The manuscript is well-written but missing some crucial components that are particularly important for the earthquake early warning, and the ML in Seismology community. I have listed a few comments below:

1. What makes EEW challenging is not the speed of the algorithm, but the trade-off between speed and accuracy. This is known as the non-determinism of earthquakes rupture, where the first few seconds of the P-wave (usually 3 seconds) do not contain any information of large earthquake's final magnitude. (see the work of Rydelek & Horiuchi, 2006 or Meier et al., 2016, 2017). This is especially true for large (M7+) earthquakes, which rupture for more than tens of seconds to minutes, and the reason why the latest EEW systems have the source tracking mechanism (i.e. update the magnitude when the source grows). However, such limitation for larger earthquakes has not explored in the manuscript. It seems that all the testing events are smaller than M6.5. I see no point in developing another EEW model for the small events, as there are already many existing approaches that can achieve similar performance. It would be more beneficial if the authors tested their model with larger earthquakes. For example, the M9.0 Tohoku, Japan earthquake or the M7.8 Turkey earthquake to see how their approach outperforms the state-of-the-art.

2. Whether the model actually generalizes across different geological settings or if it is the result of data leakage is suspecting. Seismic wave is the convolution of source, path, and site effects. Even the path term can be similar by assuming similar velocity structures, the source and site terms can be very different and vary source by source, station by station. To determine if the model actually generalizes across different factors, one test is to evaluate its performance with a different type of earthquake, such as subduction zone earthquakes. This should be a robust test considering that all the training data are onshore events.

3. From Line #134-#136, the waveforms are normalized to ML3.0. Why this normalization? The procedure is confusing and may potentially cause data leakage, a serious problem in ML. There is a relationship between magnitude, amplitude and distance, and once you normalize the amplitude to ML3.0, it implies that you already know the maximum amplitude of a waveform and the distance. However, in real-time, without having both pieces of information, how do you perform the normalization?

4. Line#157-159 show the procedure of the model, which has multiple steps including earthquake detection and locating, and finally magnitude estimation. There are extensive similar work published by other groups already. Some of them are mentioned in the introduction. To list a few there are newer methods with the generalization capability (i.e. using graph neural networks) or automatically associating multiple events (McBreaty et al., 2019; Zhu et al., 2022; McBreaty & Beroza, 2023). How does your proposed method stand out from those studies should be more clear.

Some minor comments:

Line 151-153: Recalculate the target magnitudes to fit the neural networks seems putting the cart before the house.

Line 157-159: Why 0.7 as the detection threshold? and why 0.6 for the location?

Line 181-183: Magnitude underestimation is considered a failure of an EEW model even if you can detect an event that early.

Line 209-227: For the test (Fig. S3), the magnitude estimation makes sense because, with longer data, more information is available. However, in the case of the location test, the model exhibits better performance with shorter data (~6 s) for Osaka. Conversely, for Ridgecrest, longer data (~16 s) is needed to best locate the source. I wonder if this is caused by different geological settings. If so, how does the model generalize to various regions?

Communications Earth & Environment is committed to improving transparency in authorship. As part of our efforts in this direction, we are now requesting that all authors identified as 'corresponding author' create and link their Open Researcher and Contributor Identifier (ORCID) with their account on the Manuscript Tracking System prior to acceptance. ORCID helps the scientific community achieve unambiguous attribution of all scholarly contributions. You can create and link your ORCID from the home page of the Manuscript Tracking System by clicking on 'Modify my Springer Nature account' and following the instructions in the link below. Please also inform all co-authors that they can add their ORCIDs to their accounts and that they must do so prior to acceptance.

Author Rebuttal letter: The author's response to these comments can be found at the end of this file.

Version 1:

Decision Letter:

Dear Professor Zhang,

Your manuscript titled "Universal Neural Networks for Real-Time Earthquake Early Warning Trained with Generalized Earthquakes" has now been seen by our reviewers, whose comments appear below. In light of their advice we are delighted to say that we are happy, in principle, to publish a suitably revised version in Communications Earth & Environment provided you discuss how the method performs with continuous data and clarify any limitations associated with this.

We therefore invite you to revise your paper one last time to address the remaining concerns of our reviewers. At the same time we ask that you edit your manuscript to comply with our format requirements and to maximise the accessibility and therefore the impact of your work.

EDITORIAL REQUESTS:

*****Please take care to match our formatting and policy requirements. We will check revised manuscript and return manuscripts that do not comply. Such requests will lead to delays. *****

SUBMISSION INFORMATION:

OPEN ACCESS:

Communications Earth & Environment is a fully open access journal. Articles are made freely accessible on publication. For further information about article processing charges, open access funding, and advice and support from Nature Research, please visit <https://www.nature.com/commsenv/open-access>

Link Redacted

Best regards,

Luca Dal Zilio

Editorial Board Member
Communications Earth & Environment

Joe Aslin
Deputy Editor,
Communications Earth & Environment
<https://www.nature.com/commsenv/>
Twitter: @CommsEarth

REVIEWERS' COMMENTS:

Reviewer #1 (Remarks to the Author):

The revised manuscript has addressed my previous concerns, especially by providing benchmark tests against current EEW practices over large events. I am satisfied with the revision.

I have one more concern. For demonstrating and comparing, the authors use and present results from event segments, not continuous data. For EEW, what I understand is that robustness/reliability is equally crucial as we do not want to issue many false alarms. Therefore, I am wondering how the method performs on continuous data (which most EEW systems work on), which might contain glitches, data gaps, and clipping (amplitude saturation) Would false positives be an issue for applying the proposed method in reality? What is the false positive rate when applying to real continuous data? How to tackle these data issues in your model building/training.

Reviewer #2 (Remarks to the Author):

Thank you for taking into account my previous reviews. Although I could say that it is possible to accept the manuscript in its current form, I have some minor comments below.

1. I see that the magnitude estimation results in Fig. 4 improved because you incorporated M>6 seismograms from the KIK-NET database. However, is there bias due to the type of instrument you have added? For example, what would happen if you estimate a small earthquake (M3-M4) recorded by a strong-motion accelerometer? Since you did not incorporate low magnitude seismograms recorded by accelerometers, the algorithm may overestimate small earthquakes recorded by strong-motion accelerometers.

2. Line 178-183: Is the final magnitude calculated by averaging the estimated magnitudes from single stations that contain at least 2 seconds of data, or by using the entire network of stations as input if each station has at least 2 seconds of the earthquake? Please clarify.

If one station can have more seconds of earthquake data than another, and the magnitude estimation is the average of each station's estimate, it may be that the final magnitude is underestimated due to averaging stations that have more seconds with those that have fewer seconds of the earthquake.

3. Line 181: Connect "However" better since the previous sentence already contains this word.

4. Line 313: "However, the initial magnitude result of M 5.4 indicates the arrival of a large earthquake." Not necessarily, it may be due to an earthquake of around M5.4 or even some overestimation.

5. Change K-Net to K-NET and add it to the Acknowledgments section in the format that NIED suggests (see https://www.kyoshin.bosai.go.jp/kyoshin/docs/overview_kyoshin_index_en.html).

6. What are the differences between the magnitude estimates in Fig. 5 and those in Figs. 6 and 7?

Author Rebuttal letter: The author's response to these comments can be found at the end of this file.

[revised manuscript text omitted]

**Results**

**Network models and generalized earthquakes**

We have designed three neural networks for earthquake detection, location, and magnitude
estimation, with seismic data streams continuously fed into the network models (Fig. 1). **The**
**earthquakes are simultaneously detected and located with waveforms from multiple stations, and**
**then the neural network estimates the earthquake magnitude for each station when an earthquake**
**is detected.** Our models are built upon fully convolutional networks with detailed configurations
described in the Methods section. The outputs are labeled using 1D or 3D Gaussian distributions
to represent the detected P arrival, location, and magnitude. We incorporate station XY coordinates
as two channels of the network input and sort the input data by station X and Y coordinates,
ensuring that a specified earthquake corresponds to a unique input with well-defined station order.
To ensure an universal neural network, the key technique involves the production of training
samples by recombining station seismograms to create generalized earthquakes occurring within
the study area with arbitrary station distributions (Fig. 2). Similar to the 1D layered velocity model
approximation used in common location methods^{17,39}, we adopt the assumption that seismograms
with the same epicentral distance and depth across various global areas exhibit similarity in phase
arrival times. This assumption enables us to recombine collected seismograms and generate
training earthquakes in arbitrary monitoring areas. The base training dataset comprises 94,586
single-station seismograms from real earthquakes that occurred in Italy, Oklahoma (US), and

Southern California (US) (Fig. 3). We group the 94,586 station seismograms by the corresponding
epicentral distance and earthquake depth (Fig. 2 and 3), and randomly recombine the seismograms
to create 355,001 training earthquakes. The 3D Gaussian distribution outputted by the neural
network represents an earthquake location area of $50 \text{ km} \times 100 \text{ km}$ horizontally, while the station
distribution covers the earthquake location range within a broader region of $82 \text{ km} \times 110 \text{ km}$.
Therefore, in a typical monitoring setting, we assume that the generalized stations are distributed
within a range of 0-82 km in the X direction and 0-100 km in the Y direction, while the generalized
earthquakes occur within the monitoring area of 16-66 km in the X direction, 0-100 km in the Y
direction, and 0-20 km in the Z direction. To generate a training sample, we randomly select an
earthquake location (s_x, s_y, s_z) within the earthquake range and 4-12 station locations within the
station range. We then calculate the epicentral distance r and randomly select a three-component
waveform with epicentral distance of r and depth of s_z from the grouped base dataset for a synthetic
station. Fig. 2A illustrates a representative generalized training earthquake monitored by four
stations with waveforms from two different real earthquakes in the base dataset. The E and N
components of the waveforms are rotated to the new azimuth direction centered at the generalized
earthquake location, allowing us to use phase azimuth to constrain earthquake location. To
accommodate amplitude variations among different earthquakes, we filter the three-component
seismograms within a frequency range of 1-9 Hz and normalize their amplitudes to an equivalent
magnitude of M_L 3.0. We achieve this normalization using Hutton and Boore (1987)'s empirical
equation⁴⁰: $M_L = \log(A) + 1.110 \log(r/100) + 0.00189(r-100) + 3.0$. We also generated 2000 training
earthquakes outside the monitoring areas to enhance the network's ability to distinguish abnormal
events, which are labeled as zeros. To simulate the real-time EEW process, we shift the waveforms
of the generalized earthquakes and vary the length of effective signals in the current time window

(30 s) to generate training samples when only a subset of stations is triggered (see Methods section).
The generalized earthquake data contain the crucial features related to earthquake location,
although we omit many other physical factors, such as the focal mechanism and complex
geological structures across various regions. The neural networks, trained with the generalized
earthquakes, are subsequently applied to various regions (Fig. 3).

**Pseudo real-time monitoring of the main shocks in Osaka, Japan and Ridgecrest, US**

We apply the trained neural networks to the main shock (M_{JMA} 6.1) that occurred on 18 June
2018 in Osaka, Japan and the main shock (M_w 6.4) that occurred on 4 July 2019 in Ridgecrest, US
(Fig. 4). The two earthquakes were monitored by 12 stations with different spatial distributions,
however, many of which recorded clipped waveform - where the amplitude of the seismic signal
exceeds the dynamic range of the recording instrument - potentially resulting in an underestimated
magnitude, as shown in Fig. S1. Following our magnitude estimation process, we recalculated
their magnitudes as M_L 5.5 and M_L 5.8, respectively, to facilitate consistent comparison with the
predicted results of the neural networks. To simulate real-time monitoring, the continuous
waveforms from the 12 stations are fed into the neural network with a 30 s truncating window and
0.5 s time interval along with the station locations. The detection and location networks output the
1D and 3D probability density functions (PDFs) with Gaussian distributions simultaneously. A 1D
detection label with a maximum PDF greater than 0.7 indicates an earthquake detected in the input
time window, while a 3D location label with a maximum PDF greater than 0.6 indicates a well-
located earthquake. The first triggered P arrival is identified by the maximum value in the detection
label. The origin time is calculated by subtracting the travel time from the first triggered P arrival
time. In addition, we calculate the epicentral distances and theoretical P arrival times to determine
the triggered stations and estimate their magnitudes. If the theoretical P arrival time falls within

the current time window, it indicates that the corresponding station is triggered. The magnitude
neural network takes the epicentral distances and waveforms from the triggered stations as input,
and outputs 1D Gaussian distributions that represent the normalized magnitudes which exclude
the contribution of maximum amplitudes. The M_L magnitudes are calculated as the addition of the
normalized magnitudes and the logarithm of the maximum amplitudes. The predicted magnitudes
are considered robust if the maximum PDFs exceed 0.6 and the earthquake signals last for more
than 2 seconds (estimated by the difference between the current time and the theoretical P arrival
time). **The final M_L magnitude of the detected earthquake is the mean of the triggered stations.**

[revised manuscript text omitted]

and earthquake depth. Four single-station seismograms (S1, S2, S3, S4) in panel A are extracted

from the base dataset and marked as white dots. **(C)** The magnitude statistics for the base dataset.

**Figure 3. The training data and testing sites.** The base training dataset includes 94,586 single-
station seismograms from earthquakes in Central Italy, Southern California (US), and Oklahoma
(US), indicated by blue dots; the triangles represent the stations. The red stars denote the testing
sites for earthquake sequences in Osaka, Japan, and Ridgecrest, US, while the red rectangles
indicate larger testing areas including 130 relatively large earthquakes (red dots) across Japan and
Northern California (US).

Figure 4. Real-time monitoring of two main shocks. (A) The earthquake (M_{JMA} 6.1 or M_L 5.5) occurred on June 18, 2018, in Osaka, Japan. **(B)** The earthquake (M_w 6.4 or M_L 5.8) occurred on July 4, 2019, in Ridgecrest, US. The monitoring snapshots at the 4th and 15th seconds after the P arrival of the first triggered stations are shown in the figures. The black, blue, and red curves represent the input waveforms (Z components), magnitude PDFs, and detection PDFs, respectively; the blue dashed lines mark the mean of the predicted magnitudes; the black and white stars in the right figures are the predicted and cataloged earthquake locations.

**Figure 5. The monitoring results in Osaka, Japan (A, C, E) and Ridgecrest, US (B, D, F). (
[revised manuscript text omitted]

represents a normalized magnitude. This normalization is defined by subtracting the logarithm of
the maximum amplitude from the local M_L magnitude, excluding the contribution of waveform
amplitude³⁵. This normalized magnitude allows us to standardize the neural network input with
the maximum amplitude. The relationship between waveform amplitude and magnitude is explicit,
with a tenfold increase in amplitude corresponding to a one-unit increase in magnitude. The final
prediction of the neural network must add the logarithm of the waveform amplitude to yield the
final M_L magnitude.

The neural network architectures predominantly utilize Convolutional, MaxPooling, and
Upsampling layers, as illustrated in Fig. 1. The design of the detection and location neural networks
incorporates 2D layers to handle input data from multiple stations. In contrast, the magnitude
network consists of 1D convolutional layers to process data from individual stations. The kernel
sizes for the 2D and 1D convolutional layers are uniformly set to 3×3 and 3, respectively. Zero-
padding is applied to the output of each convolutional layer to maintain the output size.

MaxPooling layers are employed to extract critical features for constraining earthquake
parameters, while Upsampling layers are used to adjust the final output size of the network model.
The task of earthquake detection is relatively straightforward compared to the location and

magnitude estimation. Thus, a fully convolutional neural network is sufficient to achieve the goal
of earthquake detection. To mitigate the issue of gradient vanishing, we introduce multiple copy
layers into the location and magnitude neural networks, as shown in Fig. 1.

The three models are trained using the Adam algorithm with a learning rate of 10^{-4} .
Additionally, they are equipped with 2, 4, and 2 Dropout layers in the detection, location, and
magnitude networks, respectively⁴⁵. The trained detection and location network models are merged
into a single file, enabling simultaneous detection and location. When the merged neural network
detects an earthquake event, it calculates the theoretical P arrival times. The triggered stations are
determined based on whether the P arrival times fall within the monitoring time window. The
waveform data from the triggered stations are then passed to the magnitude network to estimate
magnitudes, and the final result is the mean value across all triggered stations.

**References**

- 1. N.-C. Hsiao, Y.-M. Wu, T.-C. Shin, L. Zhao, T.-L. Teng, Development of earthquake early
warning system in Taiwan. *Geophysical Research Letters* 36, L00B02 (2009).
- 2. H. Nakamura, S. Horiuchi, C. Wu, S. Yamamoto, P. A. Rydelek, Evaluation of the real-time
earthquake information system in Japan. *Geophysical Research Letters* 36, L00B01 (2009).
- 3. R. M. Allen, D. Melgar, Earthquake Early Warning: Advances, Scientific Challenges, and
Societal Needs. *Annual Review of Earth and Planetary Sciences* 47, 361-388 (2019).
- 4. R. M. Allen, M. Stogaitis, Global growth of earthquake early warning. *Science* 375, 717-718
(2022).
- 5. H. S. Kuyuk, R. M. Allen, H. Brown, M. Hellweg, I. Henson, D. Neuhauser, Designing a
Network-Based Earthquake Early Warning Algorithm for California: ElarmS-2. *Bulletin of the*
*Seismological Society of America* 104, 162-173 (2013).

- 6. H. S. Kuyuk, R. M. Allen, Optimal Seismic Network Density for Earthquake Early Warning: A
Case Study from California. *Seismological Research Letters* 84, 946-954 (2013).
- 7. A. I. Chung, I. Henson, R. M. Allen, Optimizing Earthquake Early Warning Performance:
ElarmS-3. *Seismological Research Letters* 90, 727-743 (2019).
- 8. H. S. Kuyuk, R. M. Allen, A global approach to provide magnitude estimates for earthquake
early warning alerts. *Geophysical Research Letters* 40, 6329-6333 (2013).
- 9. J. Zhu, S. Li, J. Song, Magnitude Estimation for Earthquake Early Warning with Multiple
Parameter Inputs and a Support Vector Machine. *Seismological Research Letters* 93, 126-136
(2021).
- 10. H. Peng, Z. Wu, Y.-M. Wu, S. Yu, D. Zhang, W. Huang, Developing a Prototype Earthquake
Early Warning System in the Beijing Capital Region. *Seismological Research Letters* 82, 394-
403 (2011).
- 11. C. Satriano, Y.-M. Wu, A. Zollo, H. Kanamori, Earthquake early warning: Concepts, methods
and physical grounds. *Soil Dynamics and Earthquake Engineering* 31, 106-118 (2011).
- 12. A. Lomax, C. Satriano, M. Vassallo, Automatic Picker Developments and Optimization:
FilterPicker—a Robust, Broadband Picker for Real-Time Seismic Monitoring and Earthquake
Early Warning. *Seismological Research Letters* 83, 531-540 (2012).
- 13. C. Baillard, W. C. Crawford, V. Ballu, C. Hibert, A. Mangeney, An Automatic Kurtosis-Based
P- and S-Phase Picker Designed for Local Seismic Networks. *Bulletin of the Seismological
Society of America* 104, 394-409 (2013).
- 14. F. Grigoli, L. Scarabello, M. Böse, B. Weber, S. Wiemer, J. F. Clinton, Pick- and waveform-
based techniques for real-time detection of induced seismicity. *Geophysical Journal
International* 213, 868-884 (2018).

- 15. Z. Li, M. A. Meier, E. Hauksson, Z. Zhan, J. Andrews, Machine Learning Seismic Wave
Discrimination: Application to Earthquake Early Warning. *Geophysical Research Letters* 45,
4773-4779 (2018).
- 16. Z. E. Ross, M. A. Meier, E. Hauksson, T. H. Heaton, Generalized Seismic Phase Detection
with Deep Learning. *Bulletin of the Seismological Society of America* 108, 2894-2901 (2018).
- 17. M. Zhang, W. L. Ellsworth, G. C. Beroza, Rapid Earthquake Association and Location.
*Seismological Research Letters* 90, 2276-2284 (2019).
- 18. W. Zhu, G. C. Beroza, PhaseNet: a deep-neural-network-based seismic arrival-time picking
method. *Geophysical Journal International* 216, 261-273 (2019).
- 19. Z. E. Ross, Y. Yue, M.-A. Meier, E. Hauksson, T. H. Heaton, PhaseLink: A Deep Learning
Approach to Seismic Phase Association. *Journal of Geophysical Research: Solid Earth* 124, 856-
869 (2019).
- 20. S. M. Mousavi, W. L. Ellsworth, W. Zhu, L. Y. Chuang, G. C. Beroza, Earthquake
transformer—an attentive deep-learning model for simultaneous earthquake detection and phase
picking. *Nature Communications* 11, 3952 (2020).
- 21. M. Liu, M. Zhang, W. Zhu, W. L. Ellsworth, H. Li, Rapid Characterization of the July 2019
Ridgecrest, California, Earthquake Sequence From Raw Seismic Data Using Machine-Learning
Phase Picker. *Geophysical Research Letters* 47, e2019GL086189 (2020).
- 22. M. Zhang, M. Liu, T. Feng, R. Wang, W. Zhu, LOC-FLOW: An End-to-End Machine
Learning-Based High-Precision Earthquake Location Workflow. *Seismological Research*
*Letters* 93, 2426-2438 (2022).
- 23. W. Zhu, A. B. Hou, R. Yang, A. Datta, S. M. Mousavi, W. L. Ellsworth, G. C. Beroza,
QuakeFlow: a scalable machine-learning-based earthquake monitoring workflow with cloud

computing. *Geophysical Journal International* 232, 684-693 (2022).

24. S. Horiuchi, An Automatic Processing System for Broadcasting Earthquake Alarms. *Bulletin*
*of the Seismological Society of America* 95, 708-718 (2005).

25. C. Satriano, A. Lomax, A. Zollo, Real-Time Evolutionary Earthquake Location for Seismic
Early Warning. *Bulletin of the Seismological Society of America* 98, 1482-1494 (2008).

26. A. Lomax, A. Michelini, D. Jozinović, An Investigation of Rapid Earthquake Characterization
Using Single-Station Waveforms and a Convolutional Neural Network. *Seismological Research*
*Letters* 90, 517-529 (2019).

27. S. M. Mousavi, G. C. Beroza, A Machine-Learning Approach for Earthquake Magnitude
Estimation. *Geophysical Research Letters* 47, e2019GL085976 (2020).

28. S. M. Mousavi, G. C. Beroza, Bayesian-Deep-Learning Estimation of Earthquake Location
From Single-Station Observations. *IEEE Transactions on Geoscience and Remote Sensing* 58,
8211-8224 (2020).

29. T. Perol, M. Gharbi, M. Denolle, Convolutional neural network for earthquake detection and
location. *Science Advances* 4, e1700578 (2018).

30. M. Kriegerowski, G. M. Petersen, H. Vasyura-Bathke, M. Ohrnberger, A Deep Convolutional
Neural Network for Localization of Clustered Earthquakes Based on Multistation Full
Waveforms. *Seismological Research Letters* 90, 510-516 (2018).

31. H. Shen, Y. Shen, Array-Based Convolutional Neural Networks for Automatic Detection and
4D Localization of Earthquakes in Hawai'i. *Seismological Research Letters* 92, 2961-2971
(2021).

32. X. Zhang, J. Zhang, C. Yuan, S. Liu, Z. Chen, W. Li, Locating induced earthquakes with a
network of seismic stations in Oklahoma via a deep learning method. *Scientific Reports* 10,

1941 (2020).

33. M. P. A. van den Ende, J.-P. Ampuero, Automated Seismic Source Characterization Using
Deep Graph Neural Networks. *Geophysical Research Letters* 47, e2020GL088690 (2020).

34. N. A. Vinard, G. G. Drijkoningen, D. J. Verschuur, Localizing microseismic events on field
data using a U-Net-based convolutional neural network trained on synthetic data.
*GEOPHYSICS* 87, KS33-KS43 (2021).

35. X. Zhang, M. Zhang, X. Tian, Real-Time Earthquake Early Warning With Deep Learning:
Application to the 2016 M 6.0 Central Apennines, Italy Earthquake. *Geophysical Research*
*Letters* 48, 2020GL089394 (2021).

36. J. Münchmeyer, D. Bindi, U. Leser, F. Tilmann, Earthquake magnitude and location estimation
from real time seismic waveforms with a transformer network. *Geophysical Journal*
*International* 226, 1086-1104 (2021).

37. X. Zhang, W. Reichard-Flynn, M. Zhang, M. Hirn, Y. Lin, Spatiotemporal Graph
Convolutional Networks for Earthquake Source Characterization. *Journal of Geophysical*
*Research: Solid Earth* 127, e2022JB024401 (2022).

38. S. M. Mousavi, Y. Sheng, W. Zhu, G. C. Beroza, STanford EArthquake Dataset (STEAD): A
Global Data Set of Seismic Signals for AI. *IEEE Access* 7, 179464-179476 (2019).

39. G. L. Pavlis, F. Vernon, D. Harvey, D. Quinlan, The generalized earthquake-location
(GENLOC) package: an earthquake-location library. *Computers & Geosciences* 30, 1079-1091
(2004).

40. L. K. Hutton, D. M. Boore, The ML scale in Southern California. *Bulletin of the Seismological*
*Society of America* 77, 2074-2094 (1987).

41. A. Vaswani, N. Shazeer, N. Parmar, J. Uszkoreit, L. Jones, A. N. Gomez, Ł. Kaiser, I.

- Polosukhin, Attention is all you need. *Advances in neural information processing systems* 30,
(2017).
- 42. M. Böse, E. Hauksson, K. Solanki, H. Kanamori, T. H. Heaton, Real-time testing of the on-
site warning algorithm in southern California and its performance during the July 29 2008
Mw5.4 Chino Hills earthquake. *Geophysical Research Letters* 36, (2009).
- 43. C.-Y. Hsieh, W.-A. Chao, Y.-M. Wu, An Examination of the Threshold-Based Earthquake
Early Warning Approach Using a Low-Cost Seismic Network. *Seismological Research Letters*
86, 1664-1667 (2015).
- 44. Q. Kong, R. M. Allen, L. Schreier, Y.-W. Kwon, MyShake: A smartphone seismic network for
earthquake early warning and beyond. *Science Advances* 2, e1501055 (2016).
- 45. M. Abadi, P. Barham, J. Chen, Z. Chen, A. Davis, J. Dean, M. Devin, S. Ghemawat, G. Irving,
645 M. Isard, in *OSDI*. (2016), vol. 16, pp. 265-283.

**Acknowledgments**

This work was supported by National Natural Science Foundation of China Grant (No.
U2239204 to XZ), Natural Science Foundation of Jiangxi Province Grant (20224BAB211024 to
XZ), and the Natural Sciences and Engineering Research Council of Canada Discovery Grant
(RGPIN-2019-04297 to MZ).

**Data and materials availability**

Seismic data in Italy and USA were downloaded through the International Federation of
Digital Seismograph Networks (FDSN) web services from institutions such as Italy's National
Institute of Geophysics and Volcanology (INGV), Northern California Earthquake Data Center
(NCEDC), Southern California Earthquake Data Center (SCEDC). Seismic data in Japan were
downloaded from NIED Hi-net operated by National Research Institute for Earth Science and

Disaster Resilience. The earthquake catalog used in this study can be downloaded from
<http://cnt.rm.ingv.it/> (last accessed May 2023), <http://earthquake.usgs.gov/earthquakes/search/>
(last accessed May 2023), https://www.data.jma.go.jp/svd/eqev/data/bulletin/index_e.html (last
accessed May 2023). The generalized neural networks and demo examples are available at
<https://github.com/zxiong218/EEWNet> (last accessed Jan 2024).

**Author contributions**

X. Z. designed the project, developed the method, processed the data, and analyzed the results.

664 M. Z. analyzed the results. X. Z. wrote the manuscript. M. Z. revised the manuscript.

**Competing interests**

The authors have no competing interests.

Reply to the reviews of manuscript
“Universal Neural Networks for Real-Time Earthquake Early Warning Trained with Generalized Earthquakes”

by Xiong Zhang, Miao Zhang

We thank all the editors and reviewers for your insightful suggestions and comments, which greatly improved this paper. We have revised the manuscript following the suggestions and comments. We hope our revision has improved the paper to the level of satisfaction.

Reviewer #1 (Remarks to the Author):

The manuscript entitled “Universal Neural Networks for Real-Time Earthquake Early Warning Trained with Generalized Earthquakes” presents a machine learning based approach for earthquake early warning. Its primary innovation lies in devising a new framework to generate model training datasets from real earthquake recordings, ensuring that the resulting model is independent of monitoring geometries. Consequently, the trained model can be applied to earthquake monitoring arrays of different shapes and sensor quantities, thus enabling easy adaptation to various regions. The manuscript is well-written and includes various application cases. My major concerns of this manuscript are that: firstly, the proposed method should be benchmarked with existing EEW practices; and secondly, the limitations of the proposed method should be explicitly addressed in the manuscript. I listed all my concerns below:

Authors: Thanks for your valuable suggestions. In this version, we added the comparison with the traditional methods, discussed the limitation of our models.

1.To highlight the effectiveness and benefits of the proposed approach, authors should benchmark the presented approach with the current operational EEW system or best-practice. For EEW, robustness and efficiency are critical. I noticed the processing time of the proposed approach is not explicitly mentioned. The authors should address how efficiently their method can process the streaming seismic data and generate EEW-related results. For benchmarking, it is important to compare in terms of efficiency (system running time) and precision, and clearly document the parameters and/or criteria used in both methods. In this way, people can better see the advantages of your method. For example, as mentioned in the manuscript, the authors claim their method depends less on parameters and criteria selection.

Authors: We appreciate your suggestions and have made effort to address your comments. Although we could not find the exact code online, we implemented a picking-based method using similar parameter settings and trigger criteria as ElarmS. We summarized the detail parameter settings for traditional method and our neural network method in table S1. The neural network and traditional methods take an average of 0.8s and 1s, respectively, to process 30 time windows with an interval of 0.5s for an event. The neural network method was performed on a GPU (GTX 2080ti), whereas the traditional method used a CPU (Intel(R) Xeon(R) Gold 5218). Both processing times are acceptable for EEW, although it is difficult to compare them directly due to the different hardware used. In this study, we mainly focused on discussing the parameter settings and precision of the earthquake parameter solutions. We added these discussions to the main text, along with one new figure and one table in the Supplementary Materials. In addition, we also tested the models with the M 7.3 Kumamoto earthquake in Japan on April 16, 2016 (Fig.

10), and compared the monitoring process with EEW practices documented in other literature (Yuki et al., 2016).

We added the description in result section:

“We utilize data from 349 Ridgecrest events to test the traditional picking-based method and compare the results with our neural network models. To closely simulate EEW practices in ElarmS⁵⁻⁷, we adopt similar parameter settings for phase picking, location, magnitude estimation and alert criteria whenever possible (Table S1). While actual EEW operations in ElarmS are more complex, involving data transmission, event monitoring, and information dissemination, we only implement the data processing steps from truncated waveforms to earthquake parameters. The data processing workflow reflects the challenges posed by various parameter settings compared to our neural network methods. Fig. S4 shows that the mean first alarm time of the neural networks is 0.7 seconds earlier than that of the traditional method. Additionally, the neural network models yield lower errors in earthquake location and magnitude at the first alarm time. The traditional picking-based method also requires numerous parameter settings (Table S1), potentially challenging EEW efficiency and earthquake parameter accuracy compared to neural network models.”

The detail parameters in supplementary materials:

“Table S1. The parameter settings for traditional picking-base method and neural network method.

Traditional picking-based method		Neural network
Triggers	The LTA/STA time windows are set to 2s and 1s. The threshold value of the LTA/STA is 20.0. The signal to noise ratio of the waveform is above 4.0. The log(Pd) values are in the range from -5.5 to 3.5.	Detection and location PDFs are above 0.7 and 0.6. The effective signals are longer than 2 seconds and maximum PDF exceeds 0.6 for magnitude estimation.
Location	The grid sizes are 5 km and 4 km in horizontal and depth directions for grid search method. The arrival time errors at 4 stations are less than 1 second.	
Magnitude	The magnitude is computed by empirical formula: $M=1.04\log(Pd)+1.27\log(R)+5.16$. The maximum amplitudes of the waveforms within 4 seconds are used to estimate the magnitude.	
Alert criteria	At least 4 stations triggered.	

”

–

Figure S4. The comparison between the traditional picking-based method and the neural network method. (a), (c), (e), and (g) are the statistic results of first alarm time, epicentral distance errors, depth errors, and magnitude errors for traditional method. (b), (d), (f), and (h) are the corresponding statistic results for neural network method.”

2. The limitations of the proposed method are not summarized and discussed explicitly in the manuscript. For people to better understand and use the proposed approach, it is necessary to address the limitations somewhere in the manuscript.

From my point of view (correct me if I am wrong), some limitations I am aware can be:

(1) this approach adopts a global 1D velocity model. Therefore, location bias may exist for specific regions of distinct velocity models compared to the assumptions used in the training process.

(2) there seems a limitation on the size of the applied region and/or an epicenter distance limitation (as described in Lines 479, and 489).

(3) events should be located inside the monitoring array. If this is true, then this approach does not apply to large offshore events. If this is not true, it would be nice to demonstrate the proposed approach to a mega-thrust event that occurred around Japan's coast.

(4) station elevation seems not encoded into the network, thus large topography is not considered.

(5) there is a maximum number of stations that can be adopted.

Authors: Thanks for your comments. We summarized the limitations and potential improvements in the discussion according to your suggestions:

"The current model requires the station distribution to cover the monitoring areas, which may limit its application for monitoring offshore events using onshore stations. The neural network models could be further improved by extending the monitoring area and increasing the diversity of station distributions to accommodate offshore earthquake monitoring. Our generalization method provides a framework to generate training samples in arbitrary monitoring settings, given sufficient single-station waveforms. However, the current assumption is based on a global 1D layered model to recombine the waveforms, which may increase location errors for offshore earthquakes with complex velocity structures (e.g., subduction zones). Possible improvements include applying data recombination using single-station waveforms from a specified area or using exact cataloged events in that area to further fine-tune the generalized models. Additionally, incorporating station elevation as an input channel could help accommodate large topographic variations, and increasing the number of stations could support larger monitoring areas.

The earthquake depth solutions are affected by complex velocity structures and are less sensitive to arrival time features from multiple stations compared to epicentral location. Although we focus on relatively shallow earthquakes, as they are more destructive, determining the depth of deep subduction earthquakes is more challenging. A possible improvement could be the development of an extra neural network specifically for depth determination from single-station waveforms, while using multiple-station waveforms to determine the epicentral location. This approach could simplify training and allow the neural network to leverage single-station waveform features more effectively for depth determination.

The earthquake parameter solutions may be further improved by incorporating more data from different regions globally to increase the diversity of the training samples. Since small earthquakes are always more common than large earthquakes in the training set, the magnitude error for large earthquakes may be greater than for relatively small earthquakes³⁶. In this study, we augmented the large earthquake samples by randomly shifting the truncating time window for magnitude network training; however, increasing the magnitude diversity of the original samples by using data from various areas could further improve the results."

3. The authors do not mention the model is trained on broadband sensors or strong motion sensors. From my perception, it is likely broadband sensors. So how well can the trained model be generated to data recorded by strong motion sensors? For large events, strong motion sensors will not be saturated and can record signals in high quality.

Authors: Thanks for your comments. The training datasets are mostly from broadband sensors; however, we found that the magnitude underestimation may mainly result from the ML scale and lack of large earthquakes in the training set after carefully checking the results by following the reviewers' comments. We improved the magnitude network in this version and further enhanced the expression in Method section. The current models can be applied to strong motion sensors and we added an example in the manuscript as following:

"Application to a M>7.0 earthquake and its comparison with traditional EEW practice

[revised manuscript text omitted]

4. In Figure 2, S2 and S3 do not correspond in the subfigure A and B.

Authors: We modified that figure according to your suggestion.

5. In lines 157-158, the thresholds of detection and location are set to be 0.6 and 0.7 respectively. How these parameters are determined and are they universally applicable in various applicable scenarios?

Authors: To make the threshold value settings clear, we modified the description in result and discussion sections as following:

“...The maximum PDFs surpassing the given threshold values indicate an earthquake detected and located in the input time window. We set the PDF threshold values to be 0.7 and 0.6 after analyzing the predicting results by using some testing samples, and find that these values are suitable for the monitoring in different areas...”

“...Although threshold values for Gaussian distribution PDFs need to be set, the same values can be used across regions without considering station distributions and monitoring settings. We determine the threshold values for Gaussian distribution PDFs by evaluating the predicted labels of various test samples. Our findings indicate that the current settings can ensure robust event detection and accurate earthquake parameter solutions....”

6. Lines 123-125: it seems the event will always be located in X: 16-66 km and Y: 0-100 km. Does this mean the system will only monitor events inside this region? What if for a monitoring system, events may occur in arbitrary locations, for example in region X: 1-10 km or 80-100 km?

Authors: The current monitoring range are 16-66 km in the X direction and 0-100 km in the Y direction. To improve the neural network's ability to distinguish events originating outside the monitoring area, we generated 2000 training samples outside this range and labeled their outputs as zero. If an earthquake occurs outside the monitoring area, the PDFs would be low. We have modified the description of the corresponding parts:

“...We also generate an additional 2,000 earthquake samples outside the monitoring area. These are intentionally labeled with a probability density function (PDF) of zero to represent abnormal events, thereby enabling the neural network to distinguish earthquakes occurring outside the monitoring area...”

We also discussed the distant earthquakes occurred outside the monitoring area:

“...In our application, we generate 2000 earthquakes outside of the monitoring ranges as abnormal events to train the neural networks. Earthquakes outside of the monitoring areas exhibit very different features from normal events in terms of the waveforms from multiple stations. Incorporating more teleseismic waveforms for training could further enhance the efficiency of the network models...”

7. When generating training samples, the authors compile generalized earthquakes by the combination of event waveforms from different stations and earthquakes (as shown in Figure 2). During this process, why do we need to rotate the original stations to a new azimuth? What are

the criterians to perform rotations and how to determine the new azimuth angle?

Authors: When given the three-component waveforms from a specific station and earthquake, the azimuth angle, which represents the deviation of the station from the North direction relative to the epicenter, can be provided. This angle can also be estimated from the particle motions of the wave phases, such as the direct P wave. Assuming a 1D layered velocity model, the amplitude ratio of the direct P wave reflects the back azimuth angle, indicating that the waveform components contain important features related to azimuth.

To ensure that the generalized earthquakes retain this feature, we rotate the original three-component waveform according to the azimuth angle of the new generalized earthquake. Given the station location and epicenter location of the generalized earthquake, the new azimuth angle can be calculated. The rotation angle is determined as the difference between the original azimuth in the base dataset and the new azimuth of the generalized earthquake. However, we should note that this is also an approximation without consideration of the source type. We clarified this in the manuscript:

“...The E and N components of the waveforms $(d_E, d_N)^T$ are rotated to the new azimuth direction centered at the generalized earthquake location by using the equations:

$$\begin{pmatrix} d'_E \\ d'_N \end{pmatrix} = \begin{pmatrix} \cos(\delta) & \sin(\delta) \\ -\sin(\delta) & \cos(\delta) \end{pmatrix} \begin{pmatrix} d_E \\ d_N \end{pmatrix} \quad (1)$$

where δ is the difference between the original azimuth of the waveform in base dataset and new azimuth φ of the generalized earthquake. This rotation ensures that the generalized training samples retain the azimuth features in the waveform data. ...”

8. When compiling generalized earthquakes, do you use and combine stations from the same region (in your case, Italy, SC, and Oklahoma), or do you mix the stations from different regions? If you combine event waveforms from different regions, the velocity model difference may contribute to distinct P moveouts which may smear your location problem.

Authors: The underlying assumption behind our generalized earthquake approach is the global applicability of a 1D velocity model, such as the PREM model, for earthquake location. This allows us to mix data from different regions to create a comprehensive base dataset and then recombine single-station waveforms to generate training samples. However, this method can also be tailored to specific regions to generate corresponding training earthquakes with similar velocity characteristics. We clarified the velocity problem in the discussion section.

“...Our generalization method provides an opportunity to generate training samples in arbitrary monitoring settings, given sufficient single-station waveforms. However, the current assumption is based on a global 1D layered model to recombine the waveforms, which may increase location errors for offshore earthquakes with complex velocity structures (e.g., subduction zones). Possible improvements include applying data recombination using single-station waveforms from a specified area or using exact cataloged events in that area to fine-tune the generalized models. ...”

9. Fig S3 shows the epicenter and depth errors of Okasa keep increasing after ~5 seconds of receiving data. But in comparisons, the Ridgecrest case shows these errors keep decreasing until ~15 seconds of receiving data, and then gradually increase afterward. The authors attribute this contradictory phenomenon to the effect of complex scattered waves generated by geological structures. Is this explanation supported by any geological or geophysical evidence? Please discuss this in more detail to support this argument.

Authors: We modified the corresponding parts as follows:

“...This implies that the complex scattered waves are challenging to learn, as training earthquakes are generalized using simple velocity assumptions while the recombined waveforms originate from various regions. The complex latter waveforms contain abundant features unique to their local regions...”

10. Fig S4, it is better to describe the magnitudes of the two earthquakes in the figure caption.

Authors: We added the magnitude of the detected earthquake in the caption.

11. For the detection network, it is described: "The input size ($12 \times 1024 \times 3$) is determined by the maximum number of stations, the maximum length of time samples, and the number of components. Specifically, the total length of the waveform is 30 seconds, with a time interval of 0.05 seconds, resulting in 600 time samples. Any remaining 424 time samples are padded with zeroes." I am wondering why not use 600 samples directly but rather use 1024 with the remaining 424 zeros. Process less but meaningful data would be more efficient and sensible.

Authors: This arrangement results from the network structure. The decoding and encoding parts require MaxPooling and UpSampling to adjust the feature size. We set the number of time samples to be 2^n to facilitate this adjustment. We added the explanation in the method section.

“...Specifically, the total length of each waveform is 30 seconds with a time interval of 0.05 seconds, resulting in 600 time samples. To meet the requirements of MaxPooling and UpSampling, which need the time samples to be in the form of 2^n , the remaining 424 (1024-600) time samples are padded with zeroes...”

Reviewer #2 (Remarks to the Author):

Dear authors and editor, thanks for an interesting article. The paper presents an earthquake early warning algorithm that is scalable to different geographic scenarios using data recombinations and deep learning neural networks. My concern is, the earthquake early warning algorithm is more important for large earthquakes ($M > 6$), which are not included in the training database. I have some questions and comments as shown below and in the attached PDF (please open with Adobe Acrobat or Microsoft Edge) and hope that these be useful to improve the article.

Authors: Thanks for your suggestions and we sincerely appreciate this key comment. Our previous earthquake magnitudes are in ML scale in both testing and training set. We found that including large earthquakes is essential for mitigating the magnitude saturation problem. To address this, we added more large earthquakes ($M > 6$) to the training set for the magnitude network and included the maximum waveform amplitude as one of the input channels. This significantly improved our results. We updated all the results and added more large earthquakes to test the performance of our network models.

1. In Fig. 4, earthquakes of M6.1 and M6.4 reveal that the algorithm underestimates magnitudes, and the estimates do not exceed M6 for either example. This discrepancy may stem from sample saturation or the algorithm's inability to extrapolate magnitudes beyond the training range. Could you specify the type of signal recorded by the stations? Are they using broadband velocimeters or strong-motion accelerometers? More important, examining examples with unsaturated signals, such as 2011 M9 Tohoku earthquake or the 2003 M8.3 Tokachi-Oki earthquake, using public data from the K-NET and KiK-net databases, may aid in interpreting the results.

Authors: Thanks for your suggestions. We agree with you. In the previous version, we scaled the magnitude to ML in both the testing and training sets, and the training set contained too few large earthquake samples, leading to underestimation of magnitudes. In the current version, we modified the magnitude neural network's input by incorporating the normalization factor of waveforms and trained the neural network with more $M > 6.0$ earthquakes downloaded from the K-net of Japan. We updated all the magnitude results and added new tests by using a M 7.3 earthquake recorded by K-net (strong motion sensor) of Japan.

We modified the method and dataset section:

"...Since the majority of the collected earthquakes are relatively small, we included 21 $M > 6.0$ large earthquakes that occurred in Japan between October 6, 2000, and April 11, 2011, incorporating 1,454 single-station waveforms to aid in the training of the magnitude network. Although the portion of large earthquakes in the base dataset is relatively small, we enhanced their significance by data augmentation for the magnitude network training. ..."

"...For the magnitude network training, we utilize single-station waveforms as input directly. We generate 200,000 samples from the 94,586 base waveforms by randomly cutting the time windows. To emphasize the significance of large earthquakes, we generate 100,000 additional samples from the 1,454 base waveforms of $M > 6.0$ earthquakes using the same augmentation method. This ensures the model is well-trained to accurately assess the magnitude of large earthquakes..."

“...Regarding the magnitude network, we calculate the magnitude for each station individually, with an input size of 1024×5 . The first three channels consist of normalized waveform data, and the normalization factor is the maximum amplitude in m/s. The fourth channel represents the epicentral distance and the fifth channel is logarithm of the normalization factor for the waveform. Similar to the input for the detection and location neural networks, the 30-second seismic waveforms occupy 600 time samples, and any remaining 424 time samples are padded with zeros. Additionally, the epicentral distance is normalized to a range of 0-1, corresponding to distances from 0 to 110 km. Similarly, the logarithm of normalization factor is also scaled to a range of 0-1, corresponding to values from -10.0 to 0.0 ...”

We updated all the magnitude results, and the Osaka and Ridgecrest earthquakes are as following:

Figure 5. The monitoring results in Osaka, Japan (a, c, e) and Ridgecrest, US (b, d, f). (a) The predicted results of 179 cataloged earthquakes between January 16 and December 25, 2018, and 126 detected earthquakes (orange) from one-day continuous data on June 18, 2018, in Osaka, Japan. The predicted (red) and cataloged (blue) earthquake locations are connected for comparison. (b) The predicted results of 349 cataloged earthquakes between July 4, 2019 and November 16, 2020, and 389 detected earthquakes (orange) from one-day continuous data on July 4, 2019, in Ridgecrest, US. The predicted (red) and cataloged (blue) earthquake locations are connected for comparison. (c) The magnitude comparison between the cataloged and predicted results of the 179 earthquakes in Osaka, Japan. (d) The magnitude comparison between the cataloged and predicted results of the 349 earthquakes in Ridgecrest, US. (e) The magnitude statistics of the one-day monitoring in Osaka, Japan. (f) The magnitude statistics of the one-day monitoring in Ridgecrest, US.

We added the testing results for the M 7.3 Kumamoto earthquake as following:

“Application to a M>7.0 earthquake and its comparison with traditional EEW practice

We apply the trained neural network models to the M 7.3 Kumamoto earthquake in Japan on April 16, 2016. The waveforms are recorded by K-Net and converted to velocity to serve as inputs for the neural networks. Our results show that earthquake parameters are first reported at 3.5 seconds from the onset of first P arrival, with epicentral location and magnitude errors of 6.1 km and 1.4, respectively (Fig. 10a). Given that the rupture duration for large earthquakes typically exceeds 6 seconds, waveforms within 3.5 seconds may not be sufficient to determine the final earthquake magnitude. However, the initial magnitude result of M 5.4 indicates the arrival of a large earthquake. Additionally, although there are artifacts due to truncating the downloaded incomplete waveforms in Fig. 10a, our method robustly determines location and magnitude, as the M 7.3 earthquake's energy dominates the time window. By the 15th second, all stations are triggered, resulting in location and magnitude errors of 5.0 km and M 7.1, respectively (Fig. 10a). Despite most training data being velocity from broadband seismometers, our models performed well with strong motion seismometers. ...”

Figure 10. Application to the M 7.3 Kumamoto earthquake in Japan on April 16, 2016. (a) The monitoring snapshots at first alarm and 15th second from first P arrival. The black, blue, and red curves represent the input waveforms (Z components), magnitude PDFs, and detection PDFs, respectively; the blue dashed lines mark the mean of the predicted magnitudes; the black and white stars in the right figures are the predicted and cataloged earthquake locations. (b) The predicted magnitude at different time. The gray curves represent the predicted magnitudes for 12 monitoring stations, while the blue curve shows the mean magnitude. At points A, the detection and location PDFs are 0.95 and 0.60, and the system reports the first alarm. The magnitudes before point A are the results with location PDF below 0.6. At point B, the detection and location PDFs are 0.96 and 0.68, and the predicted magnitude is M 5.9 for comparison with the traditional method (Yuki et al., 2016). For the traditional method, the initial issuance time is 01:25:14.0 with a magnitude of M 5.9 (Point C). The magnitude is then updated to M 6.9 at 01:25:18.7 (Point D).

2.If the algorithm is employed in earthquake early warning scenarios, such as in South America where earthquakes in the subduction zones of Chile, Peru, or Ecuador often exceed depths of 40 km, its efficacy may be compromised. This is attributed to the shallow depth range covered in its training, prompting concerns regarding its adaptability and effectiveness in accurately estimating earthquakes at such depths.

Authors: Since the shallow earthquakes are more destructive than deep earthquakes and the majority of the collected training earthquakes are above 17.0 km, our current models focus on the earthquakes in depth above 22.8 km. However, the subduction zone may exhibit more complex velocity structures, we added more offshore earthquake tests to address this problem to

some degree. In future, the learning from deeper earthquake datasets may require further studies, we discussed it in the manuscript.

“The earthquake depth solutions are affected by complex velocity structures and are less sensitive to arrival time features from multiple stations compared to epicentral location. Although we focus on relatively shallow earthquakes, as they are more destructive, determining the depth of deep subduction earthquakes is more challenging. A possible improvement could be the development of an extra neural network specifically for depth determination from single-station waveforms, while using multiple-station waveforms to determine the epicentral location. This approach could simplify training and allow the neural network to leverage single-station waveform features more effectively for depth determination.”

We added 9 offshore earthquakes in the bellow figure, and discussed it in the manuscript.

“To test the performance of our application for offshore events with complex velocity structures, we also downloaded the event waveforms of nine earthquakes from Japan's S-net...However, the mean errors of the Japan offshore earthquakes are 7.3 km, 8.7 km, and 0.22 for epicentral location, depth, and magnitude respectively (Fig. 8), which are larger than the onshore earthquakes and which means that the complex velocity structures would affect the earthquake locations (Fig. S5). ...”

“...The current model requires the station distribution to cover the monitoring areas, which may limit its application for monitoring offshore events using onshore stations. The neural network models could be further improved by extending the monitoring area and increasing the diversity of station distributions to accommodate offshore earthquake monitoring. Our generalization method provides a framework to generate training samples in arbitrary monitoring settings, given sufficient single-station waveforms. However, the current assumption is based on a global 1D layered model to recombine the waveforms, which may increase location errors for offshore earthquakes with complex velocity structures (e.g., subduction zones). Possible improvements include applying data recombination using single-station waveforms from a specified area or using exact cataloged events in that area to further fine-tune the generalized models.... ”

Figure 8. The real-time monitoring of the neural networks to the 139 relatively large earthquakes occurred in different regions. (a) Fifty-seven \$M \geq 5.0\$ onshore earthquakes and nine \$M \geq 5.0\$ offshore earthquakes in Japan. (b) Seventy-three \$M \geq 4.0\$ earthquakes in Northern California, US. The stars are the predicted locations with the color indicating the number of monitoring stations, and the blue dots are the detected earthquakes from one-hour continuous data following the large earthquakes. The black triangles are the selected monitoring stations for the earthquakes. (c) Epicentral distance errors. (d) Depth errors. (e) The comparison between the predicted and true magnitudes. (f) The magnitude statistics of the detected earthquakes for both regions.

We also present the detail monitoring snapshots for an offshore earthquake in Supplementary materials:

Figure S5. The testing results for the M 6.0 offshore Ibaraki earthquake in Japan on August 4, 2021. The monitoring snapshots at first alarm and 15th second from first P arrival are shown in the figure. The black, blue, and red curves represent the input waveforms (Z components), magnitude PDFs, and detection PDFs, respectively; the blue dashed lines mark the mean of the predicted magnitudes; the black and white stars in the right figures are the predicted and cataloged earthquake locations. The first alarm time is 5.2 seconds after the onset of initial P arrival, with epicentral location, depth, and magnitude errors of 2.1 km, 12.3 km, and 0.4, respectively. By the 15th second, the magnitude error improves to 0.2, though the epicentral location error increases to 8.4 km. The relatively large location errors may result from complex velocity structures, as most training samples for the location network are from onshore earthquakes.

3. Considering alert messages are disseminated after station records surpass specific acceleration or magnitude thresholds, reevaluating the time of the first alarm (T_f) would be more relevant if based on when the algorithm estimates a magnitude threshold (e.g., M5 or M6 for alert issuance). This approach aligns with Fig. 9, where all earthquakes are assumed to trigger alert messages. Moreover, exploring T_f for events with magnitudes larger than M6 or M7 becomes crucial for EEWS, because the Source Time Function duration is typically more than 6 seconds (Meier et al. 2017, Science).

Authors: Thanks for your suggestions. Figure 9 demonstrates the efficiency of our neural network approach in providing robust earthquake parameter solutions. The earthquake parameters are outputted instantly when the corresponding threshold values are met. The appropriate magnitude level for public dissemination may vary across different application sites or requirements of different scenarios. To measure an algorithm's performance, the estimated magnitude should closely approach the true magnitude as quickly as possible during monitoring. To illustrate this, we present the evolution of the predicted magnitude during the monitoring of the M 7.3 Kumamoto earthquake (Figure 10b) and compare the results to the traditional method as referenced by Yuki et al. (2016). This comparison and discussion are included in the manuscript.

“... We also compare the results with the EEW practice from Yuki et al. (2016)...In traditional method, the first issuance time is 01:25:14 with magnitude of M 5.9, which is 2 seconds later than the neural network model to report the same magnitude as shown in Fig. 10b. The

magnitude is updated to M 6.9 at 01:25:18.7 for the traditional method; however the final magnitude approaches the true magnitude at 01:25:16.9 for the neural network model, which is also approximately 2 seconds earlier than tradition method (Fig. 10b) ...”

Figure 10. Application to the M 7.3 Kumamoto earthquake in Japan on April 16, 2016. (a) The The monitoring snapshots at first alarm and 15th second from first P arrival. The black, blue, and red curves represent the input waveforms (Z components), magnitude PDFs, and detection PDFs, respectively; the blue dashed lines mark the mean of the predicted magnitudes; the black and white stars in the right figures are the predicted and cataloged earthquake locations. (b) The predicted magnitude at different time. The gray curves represent the predicted magnitudes for 12 monitoring stations, while the blue curve shows the mean magnitude. At points A, the detection and location PDFs are 0.95 and 0.60, and the system reports the first alarm. The magnitudes before point A are the results with location PDF bellow 0.6. At point B, the detection and location PDFs are 0.96 and 0.68, and the predicted magnitude is M 5.9 for comparison with the traditional method (Yuki et al., 2016). For the traditional method, the initial issuance time is 01:25:14.0 with a magnitude of M 5.9 (Point C). The magnitude is then updated to M 6.9 at 01:25:18.7 (Point D).

Response to the comments from attached PDF:

Line 39-41: We added the corresponding references and modified the expression to clarify the single-station EEW. “ For EEW by single stations, traditional methods typically use the initial a few seconds of P waves to estimate the maximum amplitude of the displacement or period parameter for onsite warning ⁸⁻¹⁰. To avoid false alarms, traditional EEW practices often require one or more additional triggered stations to confirm the alert¹¹⁻¹². With the development of deep

learning, trained neural networks can maximize the information extracted from the initial P waves of single stations, efficiently detecting and predicting earthquake parameters¹³. For regional EEW by network stations (e.g., ElarmS), earthquake parameter determination usually requires data from at least four triggered stations to ensure accuracy⁵⁻⁷.”

Line 56-59: We delete the redundant sentence: “ The overarching challenge in EEW lies in rapidly and efficiently utilizing the limited available data to report earthquake parameters and minimize the "blind zone" which experiences the most significant damages.”

Line 69-71: We modified the expression. “ For single-station monitoring, Perol et al. (2018) utilize CNN to detect earthquakes and classify earthquake locations into several blocks within a specified region³⁸. Moreover, the epicentral distance, back-azimuth, and magnitude can be determined by combining properly extracted input waveform features with well-designed neural networks^{13, 35-37}. However, the back-azimuth, which relies on phase particle motion, may be affected by the noise level of the data. ”

Line 91: We added the database examples. “ ..., such as STEAD⁴⁷, INSTANCE⁴⁸, and DiTing⁴⁹,...”

Line 101-104: We perform earthquake detection and location neural networks parallel on GPUs (GTX 2080ti). If we detect the earthquake first, and then locate the earthquake, it becomes a serial program. The location program may need to wait the detection program. Although the parallel computation may require more computation memory, it save processing time. We explain it in the Method section. “Although parallel running of the detection and location networks may require more computational resources for all the real-time windows, running them in series, with localization waiting for detection, can result in increased processing time.”

Line 118-127: We adjust the writing structure of that section according to your suggestions.

Line 149-151: We added the new tests from K-net. We agree with your opinion. The underestimation is from the ML magnitude scaling. We updated all the magnitude results.

Line 170: We agree with you. We can also observe the phenomenon in the M 7.3 earthquake result in Fig. 10. To make the application straightforward, we use the mean magnitude as output, but we emphasize potential other choices in the manuscript. “...We calculate the mean magnitude from the triggered stations as the output result, and the early magnitude estimation may be even better in consideration of different portion of P waves in different triggered stations...”

Line 194: We emphasized the false and missing alarms for M>3.0 earthquakes. “The M>3.0 earthquakes predicted by the network models are all present in the original catalog, indicating no false alarms for M>3.0 earthquakes for both regions. Conversely, all M>3.0 earthquakes in the original catalog also appear in the network model's catalog for Japan data. However, for the Ridgecrest data, the network model's catalog is missing 15 out of 106 earthquakes with M>3.0, and 1 out of 16 earthquakes with M>4.0. This discrepancy is likely due to the high frequency of earthquakes occurring in close temporal proximity.”

Line 251 and Line 256-257: In this study, we focus on the algorithm efficiency and accuracy. We did not set the magnitude threshold values for alerts since different EEW application scenarios may have different responses and requirements for the lowest magnitude. However, we can compare Tf values with traditional method when the reported magnitude is the same as shown in Fig. 10b.

Line 277-278: We do not incorporate the absolute geography coordinates to the neural networks. The relative Cartesian coordinate is enough for our method. We clarify this in the method section.

“When applying the models to a monitoring region, we first set the origin of coordinates to ensure that the monitoring stations adequately cover the monitoring area. We then convert the geographical coordinates to relative Cartesian XY coordinates (km). The neural network outputs the earthquake locations in Cartesian XY coordinates, which we subsequently convert back to the corresponding geographical coordinates.”

Line 308-311: We removed the expression.

Line 333: We modified the figure Y axis according to your comments.

Line 411-412: We tested our algorithms to offshore earthquakes and focused on the depth range of 0-22.8 km since the shallow earthquakes are more destructive. More explanations seen the above comments.

Line 423: We added description. “We remove the mean and linear trend from the waveforms and eliminate the instrument responses by deconvolution.”

Line 425: The waveforms are filtered by 9 bandpass filter ranges in ElarmS (Chung et al., 2019). They use the features in different frequency ranges to distinguish teleseismograms. The teleseismograms in 3-6 HZ range are very clear seen.

Reviewer #3 (Remarks to the Author):

In this paper, the authors developed a deep learning model that can determine earthquake parameters from continuous seismic waveform in real-time. The model is generalized so that it can be applied to different regions with different network configurations and geological settings. They then demonstrate its performance in Japan and California, with location errors of 2.6-6.3 km and magnitude errors of 0.05-0.17, showing the potential of EEW application. The merit of this paper is purely engineering with the goal to shorten the time in data preprocessing, transfer learning (re-train in a different geological region), and minimal subjective decisions in real-time source parameter determinism.

The manuscript is well-written but missing some crucial components that are particularly important for the earthquake early warning, and the ML in Seismology community. I have listed a few comments below:

Authors: We sincerely appreciate your comments. To emphasize your questions about the comparisons from the other methods, regarding the speed and accuracy, we added more tests for large earthquakes, and compared the results of the M 7.3 Kumamoto earthquake in Japan on April 16, 2016 with traditional methods referenced from Yuki et al., 2016.

1. What makes EEW challenging is not the speed of the algorithm, but the trade-off between speed and accuracy. This is known as the non-determinism of earthquake rupture, where the first few seconds of the P-wave (usually 3 seconds) do not contain any information of large earthquake's final magnitude. (see the work of Rydelek & Horiuchi, 2006 or Meier et al., 2016, 2017). This is especially true for large (M7+) earthquakes, which rupture for more than tens of seconds to minutes, and the reason why the latest EEW systems have the source tracking mechanism (i.e. update the magnitude when the source grows). However, such limitation for larger earthquakes has not been explored in the manuscript. It seems that all the testing events are smaller than M6.5. I see no point in developing another EEW model for the small events, as there are already many existing approaches that can achieve similar performance. It would be more beneficial if the authors tested their model with larger earthquakes. For example, the M9.0 Tohoku, Japan earthquake or the M7.8 Turkey earthquake to see how their approach outperforms the state-of-the-art.

Authors: Thanks for your comments. We agree with your points that the long duration of the large earthquakes is truly a problem for the EEW. The final magnitude of the large earthquakes can not be predicted accurately because the available p wave length is too short. However, this phenomenon also indicates that every second at the onset of the earthquake is valuable for EEW. Our method provides a framework to leverage the limited available data to predict the earthquake parameters. Even the not-yet-triggered stations also provide valuable information to constrain the earthquake location since it means the earthquake is far from these stations.

We retrained the magnitude network models by using more large earthquakes and we updated all the magnitude results in the manuscript. In addition, we apply the model to the M 7.3 Kumamoto earthquake in Japan on April 16, 2016. The magnitude evolution during the monitoring is shown in Fig. 10. We also compared the results with the EEW practice for the same earthquake referenced from Yuki et al., 2016. The corresponding issuance time and magnitude

are shown in Table 1 of that paper. We added the figure and discussion in the manuscript:

“Application to a M>7.0 earthquake and its comparison with traditional EEW practice

We apply the trained neural network models to the M 7.3 Kumamoto earthquake in Japan on April 16, 2016. The waveforms are recorded by K-Net and converted to velocity to serve as inputs for the neural networks. Our results show that earthquake parameters are first reported at 3.5 seconds from the onset of first P arrival, with epicentral location and magnitude errors of 6.1 km and 1.4, respectively (Fig. 10a). Given that the rupture duration for large earthquakes typically exceeds 6 seconds, waveforms within 3.5 seconds may not be sufficient to determine the final earthquake magnitude. However, the initial magnitude result of M 5.4 indicates the arrival of a large earthquake. Additionally, although there are artifacts due to truncating the downloaded incomplete waveforms in Fig. 10a, our method robustly determines location and magnitude, as the M 7.3 earthquake's energy dominates the time window. By the 15th second, all stations are triggered, resulting in location and magnitude errors of 5.0 km and M 7.1, respectively (Fig. 10a). Despite most training data being velocity from broadband seismometers, our models performed well with strong motion seismometers. We also compare the results with the traditional EEW practice of the same earthquake documented in Yuki et al. (2016)⁵². Besides the K-net stations, the Hi-net stations may also be applied to help determine the hypocenter in the traditional method. Since the time required to obtain a precise hypocenter may be relatively long for the traditional method, a warning is issued based on predefined area divisions consisting of 188 areas across Japan. Although the definitions of time efficiency are different—we measure the efficiency from the onset of the first P arrival, while Yuki et al. (2016) measure from the lapse time after the detection operation of the first triggered station—we can compare the absolute issuance time. In traditional method, the first issuance time is 01:25:14 with magnitude of M 5.9, which is 2 seconds later than the neural network model to report the same magnitude as shown in Fig. 10b. The magnitude is updated to M 6.9 at 01:25:18.7 for the traditional method; however the final magnitude approaches the true magnitude at 01:25:16.9 for the neural network model, which is also approximately 2 seconds earlier than tradition method (Fig. 10b). We calculate the mean magnitude from the triggered stations as the output result, and the early magnitude estimation may be even better in consideration of different portion of P waves in different triggered stations.”

Figure 10. Application to the M 7.3 Kumamoto earthquake in Japan on April 16, 2016. (a) The monitoring snapshots at first alarm and 15th second from first P arrival. The black, blue, and red curves represent the input waveforms (Z components), magnitude PDFs, and detection PDFs, respectively; the blue dashed lines mark the mean of the predicted magnitudes; the black and white stars in the right figures are the predicted and cataloged earthquake locations. (b) The predicted magnitude at different time. The gray curves represent the predicted magnitudes for 12 monitoring stations, while the blue curve shows the mean magnitude. At points A, the detection and location PDFs are 0.95 and 0.60, and the system reports the first alarm. The magnitudes before point A are the results with location PDF below 0.6. At point B, the detection and location PDFs are 0.96 and 0.68, and the predicted magnitude is M 5.9 for comparison with the traditional method (Yuki et al., 2016). For the traditional method, the initial issuance time is 01:25:14.0 with a magnitude of M 5.9 (Point C). The magnitude is then updated to M 6.9 at 01:25:18.7 (Point D).

2. Whether the model actually generalizes across different geological settings or if it is the result of data leakage is suspecting. Seismic wave is the convolution of source, path, and site effects. Even the path term can be similar by assuming similar velocity structures, the source and site terms can be very different and vary source by source, station by station. To determine if the model actually generalizes across different factors, one test is to evaluate its performance with a different type of earthquake, such as subduction zone earthquakes. This should be a robust test considering that all the training data are onshore events.

Authors: Thanks for your comments. All testing datasets are completely independent of the training datasets. We have tested the method with numerous datasets from different regions. To address concerns about the effect of different geological settings and other factors, we further

added 9 testing events from Japan offshore events. Our test data types involve Hi-Net (broadband sensor), K-net (strong motion sensor), S-net (Ocean bottom sensor) in Japan, and CI (broadband sensor) in CA. We further discussed the problem in the manuscript.

Although the errors are relatively large for the offshore earthquakes, they are acceptable for the EEW application. We addressed errors of the nine offshore events in Figure 8 as following: “ However, the mean errors of the Japan offshore earthquakes are 7.3 km, 8.7 km, and 0.22 for epicentral location, depth, and magnitude respectively (Fig. 8), which are larger than the onshore earthquakes and which means that the complex velocity structures would affect the earthquake locations (Fig. S5).”

We also discussed the problems and potential improvements as following:

“The current model requires the station distribution to cover the monitoring areas, which may limit its application for monitoring offshore events using onshore stations. The neural network models could be further improved by extending the monitoring area and increasing the diversity of station distributions to accommodate offshore earthquake monitoring. Our generalization method provides a framework to generate training samples in arbitrary monitoring settings, given sufficient single-station waveforms. However, the current assumption is based on a global 1D layered model to recombine the waveforms, which may increase location errors for offshore earthquakes with complex velocity structures (e.g., subduction zones). Possible improvements include applying data recombination using single-station waveforms from a specified area or using exact cataloged events in that area to further fine-tune the generalized models. Additionally, incorporating station elevation as an input channel could help accommodate large topographic variations, and increasing the number of stations could support larger monitoring areas.”

Figure 8. The real-time monitoring of the neural networks to the 139 relatively large earthquakes occurred in different regions. (a) Fifty-seven \$M \geq 5.0\$ onshore earthquakes and nine \$M \geq 5.0\$ offshore earthquakes in Japan. (b) Seventy-three \$M \geq 4.0\$ earthquakes in Northern California, US. The stars are the predicted locations with the color indicating the number of monitoring stations, and the blue dots are the detected earthquakes from one-hour continuous data following the large earthquakes. The black triangles are the selected monitoring stations for the earthquakes. (c) Epicentral distance errors. (d) Depth errors. (e) The comparison between the predicted and true magnitudes. (f) The magnitude statistics of the detected earthquakes for both regions.

We also present the detail monitoring snapshots for an offshore earthquake in Supplementary materials:

Figure S5. The testing results for the M 6.0 offshore Ibaraki earthquake in Japan on August 4, 2021. The monitoring snapshots at first alarm and 15th second from first P arrival are shown in the figure. The black, blue, and red curves represent the input waveforms (Z components), magnitude PDFs, and detection PDFs, respectively; the blue dashed lines mark the mean of the predicted magnitudes; the black and white stars in the right figures are the predicted and cataloged earthquake locations. The first alarm time is 5.2 seconds after the onset of initial P arrival, with epicentral location, depth, and magnitude errors of 2.1 km, 12.3 km, and 0.4, respectively. By the 15th second, the magnitude error improves to 0.2, though the epicentral location error increases to 8.4 km. The relatively large location errors may result from complex velocity structures, as most training samples for the location network are from onshore earthquakes.

3. From Line #134-#136, the waveforms are normalized to ML3.0. Why this normalization? The procedure is confusing and may potentially cause data leakage, a serious problem in ML. There is a relationship between magnitude, amplitude and distance, and once you normalize the amplitude to ML3.0, it implies that you already know the maximum amplitude of a waveform and the distance. However, in real-time, without having both pieces of information, how do you perform the normalization?

Authors: Thanks for your comments. For the arrangement of the training data, we know the earthquakes' location and waveform information at the supervised learning stage. Since a generalized training earthquake's seismograms coming from actual earthquakes with different magnitudes, we should accommodate the waveform amplitudes to the same magnitude. The waveforms could be scaled to any magnitude, but should be the same magnitude for all the stations. This arrangement make the generalized training earthquake very similar to the real earthquake regarding the amplitude attenuation. The generalized earthquakes are then used for the **detection and location** network training.

However, when we apply the trained models to the real-time monitoring process (testing stage), the user only need to input the continuous waveforms and the corresponding station coordinates to the location neural network, then the location network output the corresponding earthquake location. After we obtain the predicted earthquake location, we calculate the epicentral distance for each station. For each station, we input the 30s' single-station waveforms and epicentral

distance to the magnitude network, and obtain the predicted magnitude for this station. The single-station waveforms are normalized to [-1, 1], and the normalization factor is also a channel of the magnitude network's input for the current version. Therefore, at the testing stage, the inputs are real earthquake data and not the pseudo generalized earthquake, no need to adjust the waveform amplitude as in the training data arrangement. When given a truncated time window (no matter noises or events), all we need to do is normalizing the waveforms to the range of [-1, 1] by using the maximum amplitude in the waveform window. The maximum amplitude is the maximum absolute value in the waveform window.

In the training stage, we clarified the corresponding parts as following:

"...Since the generalized earthquake's seismograms may be from actual earthquakes with different magnitudes, we filter the three-component seismograms within a frequency range of 2-8 Hz and normalize their amplitudes to an equivalent magnitude to accommodate inconsistent amplitudes. The single-station waveforms are scaled to the amplitude of A by using Hutton and Boore (1987)'s empirical equation⁵¹: $\log(A)=ML-1.110\log(r/100)-0.00189(r-100)-3.0$ for each station. We randomly recombine the seismograms to create 355,001 earthquakes within the monitoring areas for detection and location network training. ..."

In the real-time monitoring stage, we clarified the application process:

"The detection and location networks output the 1D and 3D probability density functions (PDFs) with Gaussian distributions simultaneously. The maximum PDFs surpassing the given threshold values indicate an earthquake detected and located in the input time window...The first triggered P arrival and predicted earthquake location are identified by the maximum values in the 1D and 3D PDFs. The epicentral distances for the monitoring stations are calculated by the predicted earthquake location. The origin time is estimated by subtracting the travel time from the first triggered P arrival time, then theoretical P arrival times of all the stations are calculated. If the theoretical P arrival time falls within the current time window, it indicates that the corresponding station is triggered. The magnitude neural network takes the epicentral distance, normalized waveforms, and the normalization factor as inputs and outputs the 1D magnitude PDF for each triggered station..."

"Regarding the magnitude network, we calculate the magnitude for each station individually, with an input size of 1024×5 . The first three channels consist of normalized waveform data, and the normalization factor is the maximum amplitude in m/s. The fourth channel represents the epicentral distance and the fifth channel is logarithm of the normalization factor for the waveform....Additionally, the epicentral distance is normalized to a range of 0-1, corresponding to distances from 0 to 110 km. Similarly, the logarithm of normalization factor is also scaled to a range of 0-1, corresponding to values from -10.0 to 0.0."

4.Line#157-159 show the procedure of the model, which has multiple steps including earthquake detection and locating, and finally magnitude estimation. There are extensive similar work published by other groups already. Some of them are mentioned in the introduction. To list a few there are newer methods with the generalization capability (i.e. using graph neural networks) or automatically associating multiple events (McBreaty et al., 2019; Zhu et al., 2022; McBreaty & Beroza, 2023). How does your proposed method stand out from those studies should be more

clear.

Authors: We added the corresponding references in the introduction. The numbers of the three papers are indicated by 30-32.

“Traditional travelttime-based earthquake monitoring involves many steps, including earthquake detection, phase picking, phase association, earthquake location and magnitude evaluation¹⁷⁻²⁶. With machine learning applied in some steps, the automatic construction of earthquake catalogs is now feasible without the need for manual intervention²⁷⁻²⁹. Especially, the development of various neural networks for phase detection and association, along with the enhanced generalization ability of these networks³⁰⁻³², significantly improves our capacity to monitor and understand seismic activity. However, as the workflow involving many steps, the error in either of the steps would potentially affect the final location and magnitude results, and they require more time to receive and analyze the earthquake signals. Unlike the automatic construction of earthquake catalogs, EEW systems require the rapid reporting of earthquake parameters at the very early stages of occurrence, often with limited information available from network stations.”

Some minor comments:

Line 151-153: Recalculate the target magnitudes to fit the neural networks seems putting the cart before the house.

Author: We modified it and found that the ML magnitude is saturated for evaluating the large earthquakes. We retrained the magnitude network by using more large earthquakes, and the results improved a lot. We updated all the magnitude results in the manuscript.

Line 157-159: Why 0.7 as the detection threshold? and why 0.6 for the location?

Author: We discussed it in the manuscript as following:

“... The maximum PDFs surpassing the given threshold values indicate an earthquake detected and located in the input time window. We set the PDF threshold values to be 0.7 and 0.6 after analyzing the predicting results by using some testing samples, and find that these values are suitable for the monitoring in different areas. ...”

Line 181-183: Magnitude underestimation is considered a failure of an EEW model even if you can detect an event that early.

Author: We agree with you. Different application scenarios and sites may have different lowest magnitude requirements for triggering the alarms. In our results, the magnitudes are M4.5 and M5.4 at 2.2 and 2.7 seconds, it may also indicate relatively large earthquakes in some application scenarios. We modified the last sentence. This indicates that the first alarms may be reported even earlier if the predicted magnitudes satisfied the lowest magnitude requirements.

Line 209-227: For the test (Fig. S3), the magnitude estimation makes sense because, with longer data, more information is available. However, in the case of the location test, the model exhibits better performance with shorter data (~6 s) for Osaka. Conversely, for Ridgecrest, longer data (~16 s) is needed to best locate the source. I wonder if this is caused by different geological settings. If so, how does the model generalize to various regions?

Author: It may be the reason. However, the location errors for both areas are acceptable for EEW, and more data from various regions may help to improve the results. We discussed it in the manuscript:

“Our generalization method provides a framework to generate training samples in arbitrary

monitoring settings, given sufficient single-station waveforms. However, the current assumption is based on a global 1D layered model to recombine the waveforms, which may increase location errors for offshore earthquakes with complex velocity structures (e.g., subduction zones). Possible improvements include applying data recombination using single-station waveforms from a specified area or using exact cataloged events in that area to further fine-tune the generalized models. "

Reviewer #1 (Remarks to the Author):

The revised manuscript has addressed my previous concerns, especially by providing benchmark tests against current EEW practices over large events. I am satisfied with the revision.

I have one more concern. For demonstrating and comparing, the authors use and present results from event segments, not continuous data. For EEW, what I understand is that robustness/reliability is equally crucial as we do not want to issue many false alarms. Therefore, I am wondering how the method performs on continuous data (which most EEW systems work on), which might contain glitches, data gaps, and clipping (amplitude saturation) Would false positives be an issue for applying the proposed method in reality? What is the false positive rate when applying to real continuous data? How to tackle these data issues in your model building/training.

Authors: Thanks for your comments. We tested our models for the continuous data without considering the data transmitting process. In our downloaded one-hour testing datasets, there are no false alarms for the $M > 3.0$ earthquakes. However, for the real continuous data, we discussed the false alarms related to abnormal signals and possible improvements for future studies.

“ Our models are tested using continuous data without considering network transmission, and the data may contain glitches, gaps, and clipping. The abnormal signals (e.g., data gaps, spike signals) in the noise waveforms are easily distinguished by the neural network, and using more such abnormal signals as training samples could effectively avoid false alarms (mistaking noise for earthquake events). However, if abnormal signals appear in the earthquake waveforms, they may affect the accuracy of the earthquake parameters. For example, small earthquakes with large-amplitude spike signals may be predicted as large earthquakes, leading to false alarms. Although our models are applicable for clipping data from broadband seismographs (e.g., M 6.4 Ridgecrest earthquake), performances could be further improved by incorporating more training samples with abnormal signals from real-time waveform streams.”

Reviewer #2 (Remarks to the Author):

Thank you for taking into account my previous reviews. Although I could say that it is possible to accept the manuscript in its current form, I have some minor comments below.

1. I see that the magnitude estimation results in Fig. 4 improved because you incorporated $M > 6$ seismograms from the KIK-NET database. However, is there bias due to the type of instrument you have added? For example, what would happen if you estimate a small earthquake ($M3-M4$) recorded by a strong-motion accelerometer? Since you did not incorporate low magnitude seismograms recorded by accelerometers, the algorithm may overestimate small earthquakes recorded by strong-motion accelerometers.

Authors: Thanks for your comments. I agree with you. We modified the discussion as following:
The earthquake parameter solutions may be further improved by incorporating more data from different regions and types of instruments globally to increase the diversity of the training

samples. Since small earthquakes are always more common than large earthquakes in the training set, the magnitude error for large earthquakes may be greater than for relatively small earthquakes³⁶. Additionally, large earthquakes are mainly recorded by strong-motion accelerometers, while small earthquakes are recorded by broadband seismographs in our datasets. Including more strong-motion data across various magnitudes may further improve magnitude accuracy. In this study, we augmented the large earthquake samples by randomly shifting the truncating time window for magnitude network training; however, increasing the magnitude diversity of the original samples by using data from various areas could further enhance the generalization ability.

2. Line 178-183: Is the final magnitude calculated by averaging the estimated magnitudes from single stations that contain at least 2 seconds of data, or by using the entire network of stations as input if each station has at least 2 seconds of the earthquake? Please clarify.

If one station can have more seconds of earthquake data than another, and the magnitude estimation is the average of each station's estimate, it may be that the final magnitude is underestimated due to averaging stations that have more seconds with those that have fewer seconds of the earthquake.

Authors: Thanks for your comments. We predict the magnitude for each station respectively, and then average the magnitude results of the stations with at least 2 seconds. We clarified the corresponding parts:

"The magnitude neural network predicts the magnitude by taking the epicentral distance, normalized waveforms, and the normalization factor of each single station as inputs, and outputs the 1D magnitude PDF for each triggered station."

I agree with you that the final magnitude could be further improved by increasing the weights of the stations with more effective signals. We mentioned this in the manuscript:

"We calculate the mean magnitude from the triggered stations as the final result, and the early magnitude estimation may be even better by weighted averaging the magnitudes in consideration of different signal lengths in different triggered stations."

3. Line 181: Connect "However" better since the previous sentence already contains this word.

Authors: We modified the sentence.

4. Line 313: "However, the initial magnitude result of M 5.4 indicates the arrival of a large earthquake." Not necessarily, it may be due to an earthquake of around M5.4 or even some overestimation.

Authors: Generally, earthquake magnitudes tend to be underestimated when using a small amount of data. Therefore, if the estimated magnitudes are relatively large despite the use of limited data inputs, this may warrant closer attention to the event. However, your points are also valid. We have revised the sentence as follows:

However, the initial magnitude result of M 5.4 could potentially indicate the arrival of a large earthquake, as the results are reported based on a small amount of signal data.

5. Change K-Net to K-NET and add it to the Acknowledgments section in the format that NIED suggests (see https://www.kyoshin.bosai.go.jp/kyoshin/docs/overview_kyoshin_index_en.html).

Authors: We modified the corresponding parts.

6. What are the differences between the magnitude estimates in Fig. 5 and those in Figs. 6 and 7?

Authors: The magnitudes in Fig. 5 represent the overall results when sliding the monitoring windows from the first alarms to the time when all stations received sufficient data. The final results are determined by choosing the results with the best location PDF and the effective signals occur within the range of 15~20 s, considering a reasonable length of waveforms for better parameter accuracy (as described in the manuscript). Figs. 6 and 7 show the results at specific moments when some stations may not have triggered or received sufficient data. We have clarified the corresponding sections accordingly.

Fig. 6 and 7 show comparisons between the predicted and cataloged earthquake parameters at the 4th and 15th seconds after the first triggered stations. In the comparison, we do not consider the PDF threshold values and only analyze the magnitude and location results predicted at the specified moments.